# BRCA2-DSS1 interaction is dispensable for RAD51 recruitment at replication-induced and meiotic DNA double strand breaks

Arun Prakash Mishra[1], Suzanne A. Hartford[1,5], Sounak Sahu[1], Kimberly Klarmann[1,6], Rajani Kant Chittela[1,7], Kajal Biswas[1], Albert B. Jeon[2], Betty K. Martin[1,3], Sandra Burkett[1], Eileen Southon[1], Susan Reid[1], Mary E. Albaugh[1,3], Baktiar Karim[2], Lino Tessarollo[1], Jonathan R. Keller[1,4] & Shyam K. Sharan[1✉]

The interaction between tumor suppressor BRCA2 and DSS1 is essential for RAD51 recruitment and repair of DNA double stand breaks (DSBs) by homologous recombination (HR). We have generated mice with a leucine to proline substitution at position 2431 of BRCA2, which disrupts this interaction. Although a significant number of mutant mice die during embryogenesis, some homozygous and hemizygous mutant mice undergo normal postnatal development. Despite lack of radiation induced RAD51 foci formation and a severe HR defect in somatic cells, mutant mice are fertile and exhibit normal RAD51 recruitment during meiosis. We hypothesize that the presence of homologous chromosomes in close proximity during early prophase I may compensate for the defect in BRCA2-DSS1 interaction. We show the restoration of RAD51 foci in mutant cells when Topoisomerase I inhibitor-induced single strand breaks are converted into DSBs during DNA replication. We also partially rescue the HR defect by tethering the donor DNA to the site of DSBs using streptavidin-fused Cas9. Our findings demonstrate that the BRCA2-DSS1 complex is dispensable for RAD51 loading when the homologous DNA is close to the DSB.

[1] Mouse Cancer Genetics Program, Center for Cancer Research, National Cancer Institute, National Institutes of Health, Frederick, MD, USA. [2] Molecular Histopathology Laboratory, Leidos Biomedical Research, Inc. Frederick National Laboratory for Cancer Research, Frederick, MD, USA. [3] Laboratory Animal Sciences Program, Leidos Biomedical Research, Inc. Frederick National Laboratory for Cancer Research, Frederick, MD, USA. [4] Basic Science Program, Leidos Biomedical Research, Inc. Frederick National Laboratory for Cancer Research, Frederick, MD, USA. [5] Present address: Regeneron Pharmaceuticals, Inc, Tarrytown, NY, USA. [6] Present address: Developmental Therapeutics Program, Division of Cancer Treatment and Diagnosis, NCI, Frederick, MD, USA. [7] Present address: Applied Genomics Section, Bhabha Atomic Research Center, Trombay, Mumbai, India. ✉email: sharans@mail.nih.gov

Dividing cells are subjected to a myriad of factors that cause DNA damage and genomic instability[1]. DNA double-strand breaks (DSBs) are the most dangerous of such damages. Cells have developed several mechanisms to maintain genomic integrity and repair the damaged DNA. The error-prone classical non-homologous end joining (cNHEJ) and error-proof homologous recombination (HR) are the most prominent of these mechanisms[2]. In mammalian cells, cNHEJ occurs during most phases of the cell cycle and is the predominant pathway for DSB repair in which broken ends are ligated together with minimum reference to the original DNA sequence[3]. In contrast, sequence homology between the damaged and template DNA is required for HR[4]. HR primarily occurs during the S and G2 phases of the cell cycle when the sister chromatids are present and serve as a template for the repair of the damaged DNA[5].

Repair of DSBs by HR is initiated by the recruitment of the MRN complex (MRE11–RAD50–NBS1) along with CtIP at the breaks[5]. These proteins, together with BRCA1 and BARD1, initiate the process of end resection, which generates 3′ over-hangs. The 3′ overhangs are coated with replication protein A (RPA), which stabilizes the single-stranded DNA (ssDNA)[6]. BRCA2 recruits RAD51[2,4], a key DNA repair protein, to the DSBs replacing RPA nucleofilament[4]. Displacement of RPA with RAD51 initiates strand invasion towards a homologous template, making a D-loop[4]. DNA polymerase δ (Pol δ) extends the invaded strand, utilizing the homologous DNA as a template. After the DNA is copied from the template, Holliday junctions are formed, then resolved either with a crossover (mostly in meiosis) or a non-crossover[4]. Apart from its role in HR, BRCA2 is also required for stalled replication fork stability[7] and resolu-tion of R-loops[8–10].

Inherited mutation in *BRCA2* increases the risk of female breast cancer by 69% and ovarian cancer by 17%[11]. The risk of developing other types of cancers, such as fallopian tube cancer[12], peritoneal cancer[13], pancreatic cancer, and prostate cancer[14], is also increased in *BRCA2* mutation carriers. Mutations in *BRCA2* are also associated with Fanconi anemia (FA), a rare autosomal-recessive disorder associated with childhood solid tumors and acute myeloid leukemia; consequently, *BRCA2* is also considered to be one of the 22 FA genes, *FANCD1*[15]. FA is characterized by defective DNA repair and marked increase in chromosomal instability and hypersensitivity to mitomycin C (MMC)[16]. In contrast to the inheritance of a single mutant allele of *BRCA2* in individuals who develop breast, ovarian, and other solid cancers, the inheritance of biallelic mutations in *BRCA2*[17] causes FA (D1 subtype). We have previously used a mouse embryonic stem cells (mESCs)-based functional assay to characterize a number of FA-associated BRCA2 variants and found many of them to be hypomorphic[18–20]. One such variant is c.7757T > C, which changes a leucine residue at position 2510 to proline (p.Leu2510Pro, referred to as L2510P) (Fig. 1A). Functional characterization of L2510P variant revealed a significant reduc-tion in mESC viability. Furthermore, the cells exhibited severe reduction in IR-induced RAD51 foci formation, HR, and sig-nificant increase in genomic instability[18]. Biochemical and structural analyses revealed that the change in leucine to proline (at human position 2510/mouse position 2431) disrupted the DSS1-binding site of BRCA2, thereby reducing their physical interaction[18]. DSS1 (*Deleted in split hand/split foot protein 1*), a phylogenetically conserved protein, consists of 70 amino acids and is characterized as an intrinsically disordered protein (IDP) due to its lack of a defined 3D structure (based on NMR studies) and its sequence properties[21–23]. DSS1, also known as SEM1 or SHFM1, is a 26S proteasomal complex subunit that mediates protein homeostasis[23,24]. Besides playing a crucial role in the proteasomal complex, it is also a part of other complexes, such as ubiquitin binding[24], mRNA export machinery TREX-2[25], and small RNA processing[26]. A recent study has elucidated that DSS1 also interacts with RAD52 to promote DSB repair[27]. Interaction between DSS1 and BRCA2 is considered to be critical for the stability of BRCA2[28]. Recent studies have shown that the BRCA2-DSS1 complex has a role in RPA32 removal from the DSBs[2,29]. DSS1 has also been implicated in nuclear import of BRCA2; consequently, BRCA2 variants defective in interaction with DSS1 are not localized in the nucleus, rendering them defective in HR[30].

The BRCA2 L2510P variant was observed in a family that had two live births[31]. Along with the L2510P variant, these children inherited a truncating mutation, p.Glu1550Ter (E1550X). One of the siblings was born with stage II Wilms tumor and succumbed to the treatment within 1 year. The other sibling developed T-cell acute lymphoblastic leukemia by the age of five[31]. Given the significant impact of the L2510P variant on the viability of mESCs and the significant reduction in HR, it is surprising how such a deleterious variant can even support viability in humans. To address this and examine the physiological significance of BRCA2-DSS1 interaction, we generated a knock-in mouse model (*Brca2^{L2431P}*) with a leucine-to-proline change in codon 2431 (CTG > CCA, Supplementary Fig. 1), which is equivalent to residue 2510 of the human protein[32]. Although the majority of homozygous and hemizygous mice die during embryogenesis, some are viable, suggesting that the variant can support viability. Notably, mice that are born alive are fully fertile and do not exhibit increased susceptibility to tumor formation. While embryonic and adult fibroblasts (AFs) are defective in radiation-induced RAD51 foci formation, meiocytes exhibit no defect in RAD51 recruitment during prophase I. Our findings reveal that BRCA2-DSS1 interaction is dispensable for RAD51 recruitment to the replication-induced and meiotic double-strand breaks.

## Results

**Brca2^{L2431P/ L2431P} and Brca2^{L2431P/ KO} mice are born at sub-Mendelian ratio.** To examine the physiological impact of the BRCA2 L2510P variant, we generated mice expressing an L2431P variant. We mutated codon 2431 of *Brca2* in exon 15 from CTG > CCA in mESCs by gene targeting (Fig. 1a and Supple-mentary Fig. 1a–c). We used the correctly targeted mESC clones to generate *Brca2^{L2431P/+}* mice. These mice are viable and fertile. To obtain homozygous mutants, we intercrossed *Brca2^{L2431P/+}* mice. Viable homozygous (*Brca2^{L2431P/L2431P}*, for simplicity referred to as *Brca2^{LP/LP}*) mice were born in sub-Mendelian ratio both in C57BL/6J X 129Sv/Ev mixed and pure C57BL6/J genetic backgrounds (Table 1).

FA patients with the BRCA2 L2510P variant were reported to be compound heterozygotes with a truncating mutation in the other *BRCA2* allele[31]. To mimic this compound heterozygous condition in the mouse model, we crossed *Brca2^{LP/+}* mice with mice heterozygous for the *Brca2* null allele (*Brca2^{KO/+}*) to obtain hemizygous *Brca2^{LP/KO}* mice[33]. The viability of *Brca2^{LP/KO}* mice was significantly reduced, and this reduction was more pro-nounced than what was observed in the *Brca2^{LP/LP}* mice (Table 1, $p = 3.79497E-10$ in mixed background and $p = 7.5002E-15$ in pure line). We dissected mutant embryos at various gestational days to determine the embryonic stage of lethality. We obtained both *Brca2^{LP/KO}* and *Brca2^{LP/LP}* embryos in expected Mendelian ratios at 9.5dpc, 13.5dpc, and 14.5dpc (Table 2). The number of *Brca2^{LP/KO}* embryos was significantly reduced at 16.5dpc (Table 2, $p = 0.013$). Other than a reduction in the size of the *Brca2^{LP/KO}* embryos and some of the *Brca2^{LP/LP}* embryos, we did not observe any apparent developmental defects (Supplementary Fig. 2a).

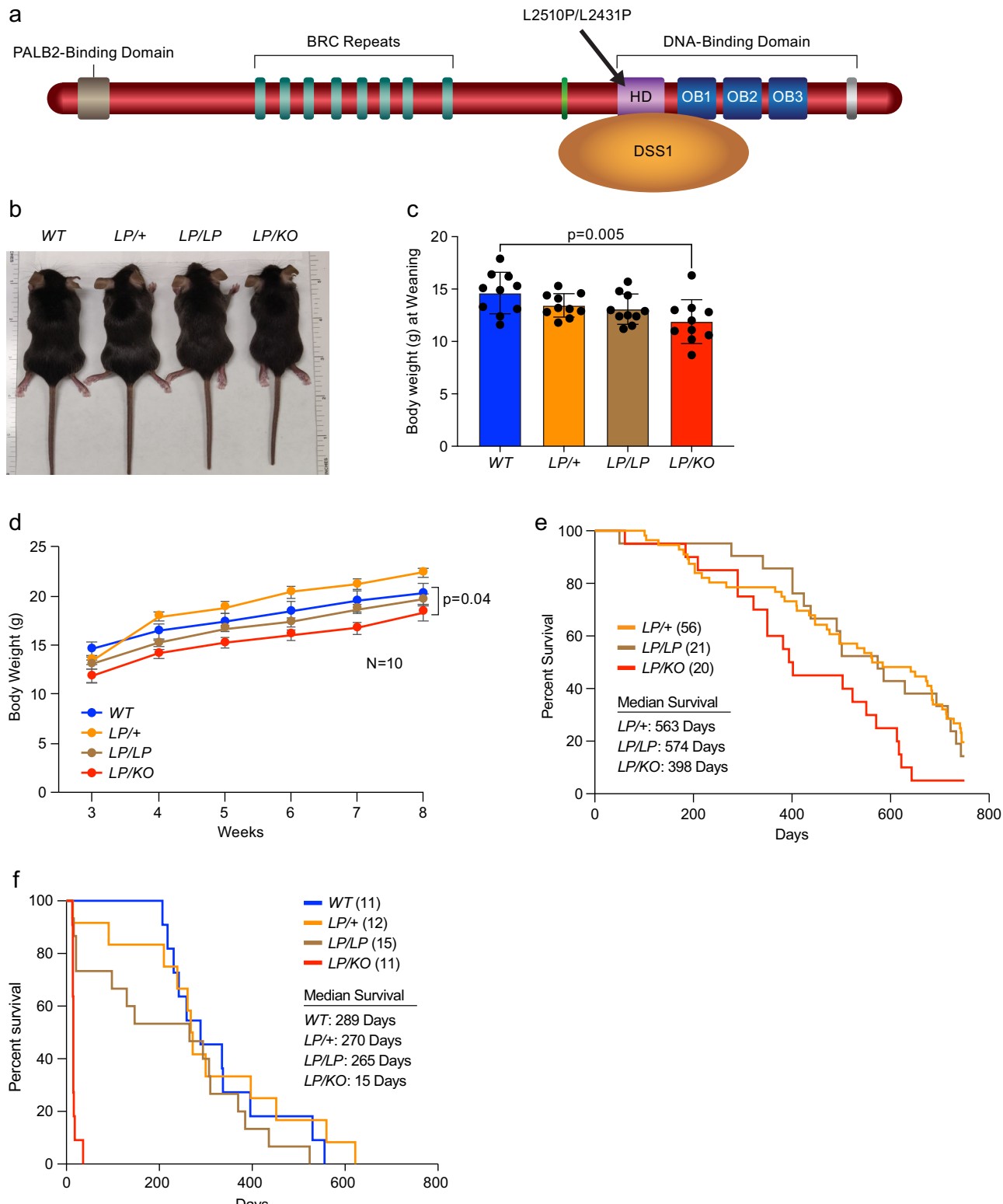

**Fig. 1 Physiology of *Brca2^L2431P* homozygous and hemizygous mice. a** Schematic representation of BRCA2 protein showing different functional domains: N-terminal PALB2 binding domain, eight BRC repeats in the middle and a C-terminal DNA binding domain consisting of a helical domain (HD) and three oligo-binding (OB) domains. BRCA2 L2520P/L2431P variant is in the HD and disrupts interaction with DSS1. **b** Representative images of mice of indicated genotypes at weaning (21 days). **c** Body weights of mice at weaning showing significantly reduced body weight of *LP/KO* mice compared to WT mice (*n* = 10, two-tailed Student's *t* test, error bar- SD of mean). **d** Body weights of mice of indicated genotypes measured for 8 weeks post-weaning, *LP/KO* mice weighed less compared to the WT littermates (*n* = 10, one-tailed *t*-test: two-sample assuming unequal variances, error bar- SE of mean). **e** Kaplan–Meier survival curve for 800 days of mice of indicated genotypes. *LP/KO* mice have reduced median survival (*p* = 0.009, Log-rank Mantel-Cox test) compared to *LP/* + and *LP/LP*. **f** Kaplan–Meier survival curve of mice of different genotypes after 8 Gy of γ-radiation, *LP/KO* mice showed median survival of 15 days and are extremely sensitive to radiation as compared to all other genotypes (*p* = 0.0001, Log-rank Mantel–Cox test).

**Table 1 Observed and expected number of offspring of various genotypes obtained from Brca2$^{LP/+}$ intercross and Brca2$^{LP/+}$ X Brca2$^{KO/+}$ cross.**

| | | LP/+ X LP/+ | | | | LP/+ X KO/+ | | | |
|---|---|---|---|---|---|---|---|---|---|
| | | +/+ | LP/+ | LP/LP | | +/+ | LP/+ | KO/+ | LP/KO |
| Mixed line | Expected | 16 | 32 | 16 | Expected | 39 | 39 | 39 | 39 |
| | Observed | 15 | 44 | 5 | Observed | 58 | 44 | 51 | 3 |
| | $\chi^2$ p-value | 0.002 | | | $\chi^2$ p-value | 3.79497E-10 | | | |
| C57Bl/6J pure line | Expected | 48.75 | 97.5 | 48.75 | Expected | 62.75 | 62.75 | 62.75 | 62.75 |
| | Observed | 65 | 106 | 24 | Observed | 103 | 79 | 55 | 14 |
| | $\chi^2$ p-value | 8.59677E-05 | | | $\chi^2$ p-value | 7.5002E-15 | | | |

**Table 2 Observed and expected number of embryos of various genotypes obtained from Brca2$^{LP/+}$ intercross and Brca2$^{LP/+}$ X Brca2$^{KO/+}$ cross at different days of gestation.**

| | | LP/+ X LP/+ | | | | LP/+ X KO/+ | | | |
|---|---|---|---|---|---|---|---|---|---|
| | | +/+ | LP/+ | LP/LP | | +/+ | LP/+ | KO/+ | LP/KO |
| 9.5dpc | Expected | 7 | 14 | 7 | Expected | 9.5 | 9.5 | 9.5 | 9.5 |
| | Observed | 9 | 11 | 8 | Observed | 9 | 8 | 15 | 6 |
| | $\chi^2$ p-value | 0.5 | – | – | $\chi^2$ p-value | 0.09 | – | – | – |
| 13.5dpc | Expected | 14 | 28 | 14 | Expected | 15.5 | 15.5 | 15.5 | 15.5 |
| | Observed | 15 | 21 | 20 | Observed | 13 | 18 | 12 | 19 |
| | $\chi^2$ p-value | 0.11 | – | – | $\chi^2$ p-value | 0.49 | – | – | – |
| 14.5dpc | Expected | 11 | 22 | 11 | Expected | 9.75 | 9.75 | 9.75 | 9.75 |
| | Observed | 9 | 23 | 12 | Observed | 11 | 10 | 13 | 5 |
| | $\chi^2$ p-value | 0.75 | – | – | $\chi^2$ p-value | 0.14 | – | – | – |
| 16.5dpc | Expected | 32.75 | 65.5 | 32.75 | Expected | 19.25 | 19.25 | 19.25 | 19.25 |
| | Observed | 28 | 78 | 25 | Observed | 20 | 30 | 16 | 11 |
| | $\chi^2$ p-value | 0.073 | – | – | $\chi^2$ p-value | 0.013 | – | – | – |
| 18.5dpc | Expected | 10 | 20 | 10 | – | – | – | – | – |
| | Observed | 12 | 18 | 10 | – | – | – | – | – |
| | $\chi^2$ p-value | 0.75 | – | – | – | – | – | – | – |
| At birth | Expected | 23.75 | 47.5 | 23.75 | – | – | – | – | – |
| | Observed | 29 | 55 | 11 | – | – | – | – | – |
| | $\chi^2$ p-value | 0.0002 | – | – | – | – | – | – | – |

Histopathological analysis also did not reveal any defects (Supplementary Fig. 2c).

When we examined the Brca2$^{LP/LP}$ embryos at 16.5dpc and 18.5dpc, they were obtained at expected Mendelian ratios. Some Brca2$^{LP/LP}$ embryos were clearly smaller in size, but we observed no other defects (Supplementary Fig. 2b). Notably, when we genotyped the newborn pups, we found the number of Brca2$^{LP/LP}$ pups to be significantly reduced ($p = 0.0002$, Table 2). We predict that the smaller embryos (at 16.5dpc and 18.5dpc) are unable to survive as we did not notice any difference in the size of the Brca2$^{LP/LP}$ pups at birth compared to their littermates. It is possible that the smaller Brca2$^{LP/LP}$ embryos were stillborn and/or were eaten immediately by their mothers.

**Brca2$^{LP/KO}$ mice are not tumor prone but are hypersensitive to radiation.** At weaning, Brca2$^{LP/KO}$ mice are smaller in size and weigh significantly less than the littermates of other genotypes (Fig. 1b, c). Although they grew normally, they remained marginally smaller compared to the mice of other genotypes (Fig. 1d). Unlike Brca2$^{LP/KO}$ mice, Brca2$^{LP/LP}$ were similar in size compared to the control littermates. Both Brca2$^{LP/LP}$ and Brca2$^{LP/KO}$ animals are fertile and produce offspring with normal litter size (Supplementary Table 1). We observed a reduction in litter size when homozygous and hemizygous animals were intercrossed. However, this is due to the lethality of Brca2$^{KO/KO}$ and some of the Brca2$^{LP/LP}$ and Brca2$^{LP/KO}$ embryos (Supplementary Table 1).

When we aged a cohort of Brca2$^{LP/LP}$, Brca2$^{LP/KO}$, and Brca2$^{LP/+}$ mice, we did not observe a significant difference in the survival of Brca2$^{LP/+}$ and Brca2$^{LP/LP}$ mice. However, Brca2$^{LP/KO}$ mice had significantly reduced survival compared to the other two genotypic groups (median survival 398 days, $p = 0.009$) (Fig. 1e). Mice showing signs of distress were euthanized and sent for histopathological analysis. Despite their shorter lifespan, Brca2$^{LP/KO}$ mice did not show increased tumor incidence. Moreover, the tumors harvested from Brca2$^{LP/LP}$, Brca2$^{LP/KO}$, and Brca2$^{LP/+}$ mice displayed a similar tumor spectrum (Table 3). Pathological analyses of Brca2$^{LP/KO}$ mice did not reveal the cause of their shorter lifespan. Brca2$^{LP/LP}$ and Brca2$^{LP/KO}$ mice did not exhibit any other overt phenotypic defect, except for a stark decrease in the mammary branching and terminal end bud (TEB) formation in Brca2$^{LP/KO}$ females (4–6 weeks old) as compared to wild-type (WT) littermates (Supplementary Fig. 2d, e). However, this did not affect their ability to nurse. The mammary gland phenotype may be due to subtle proliferation defect in adult stem cells that leads to a reduction in mammary branching and TEBs in the hemizygous mice. Similar subtle defects in other tissues could be a plausible reason for shorter life span and reduced body weights of Brca2$^{LP/KO}$ mice.

Based on the significant reduction in the number of viable Brca2$^{LP/LP}$ and Brca2$^{LP/KO}$ mice, it is evident that the L2431P variant has a clear impact on the functions of BRCA2, similar to the effect previously observed in Brca2$^{KO/KO}$ mESC expressing human the BRCA2 L2510P variant[18]. Surprisingly, we did not

**Table 3 Tumor incidence and spectrum of $Brca2^{LP/+}$, $Brca2^{LP/LP}$, and $Brca2^{LP/KO}$ mice.**

| Genotype | $Brca2^{LP/+}$ | | $Brca2^{LP/LP}$ | | $Brca2^{LP/KO}$ | |
|---|---|---|---|---|---|---|
| # of mice | 34 | | 18 | | 18 | |
| # of mice with all neoplasm | 15 (44.12%) | | 11 (61.11%) | | 7 (38.89%) | |
| | **Females** | **Males** | **Females** | **Males** | **Females** | **Males** |
| # of mice | 19 | 15 | 7 | 11 | 9 | 9 |
| Hematopoietic neoplasm | 7 | 6 | 5 | 5 | 4 | 3 |
| Osteosarcoma | 1 | – | – | – | – | – |
| Sarcoma, NOS | – | – | 1 | – | – | – |
| Hemangiosarcoma | – | 1 | – | – | – | – |
| Carcinoma | 2 | – | 2 | – | – | – |
| Hemangioma | 1 | – | – | – | – | – |
| Pituitary adenoma | – | – | 1 | – | – | – |
| Schwannoma | – | 1 | – | – | – | – |

NOS not otherwise specified.

observe any effect on fertility and/or tumor susceptibility in the $Brca2^{LP/LP}$ and $Brca2^{LP/KO}$ mice that were born alive. To examine whether the viable mice are "escapers" that somehow evaded the functional impact of the L2431P variant, we challenged these mice with 8 Gy of γ-irradiation. We found $Brca2^{LP/KO}$ mice to be hypersensitive to irradiation, as all the mice succumbed within 42 days of treatment (Fig. 1f). However, $Brca2^{LP/LP}$ mice exhibited a similar response to radiation as shown by $Brca2^{LP/+}$ and $Brca2^{+/+}$ controls. The radiation hypersensitivity among $Brca2^{LP/KO}$ mice clearly demonstrates that these mice are defective in the DSB repair function of BRCA2.

**Hematopoietic reconstitution ability of mutant progenitors.** Individuals with BRCA2 L2510P variant were reported to develop FA, which is characterized by a pronounced hematopoietic defect[31]. Therefore, we examined the hematopoietic reconstitution ability of the progenitors in $Brca2^{LP/LP}$ and $Brca2^{LP/KO}$ mice. Since the liver is the site of hematopoiesis during embryogenesis, we used liver cells of embryos (of all genotypes) that appeared normal at 16.5dpc to examine the proliferative potential of their hematopoietic progenitors. Fetal liver cells from WT and control ($Brca2^{KO/+}$, $Brca2^{LP/+}$) embryos formed round and compact colonies under culture conditions. In contrast, fetal liver cells from $Brca2^{LP/LP}$ embryos formed smaller and fewer colonies (Fig. 2a, b). These numbers were further reduced for $Brca2^{LP/KO}$ fetal liver cells (Fig. 2a, b). The observations suggest that both $Brca2^{LP/LP}$ and $Brca2^{LP/KO}$ embryos have fewer hematopoietic progenitors and have a marked defect in their proliferative ability.

Next, we examined whether the mutant fetal liver cells are proficient in hematopoietic reconstitution in mice that are lethally irradiated (10 Gy), to deplete the host hematopoietic cells. We harvested fetal liver cells from 16.5dpc embryos and transplanted them into lethally irradiated congenic recipients and monitored them for the hematopoietic reconstitution based on their survival. We observed that the mice that received $Brca2^{LP/KO}$ fetal liver cells showed significantly reduced survival compared to those receiving fetal liver cells from $Brca2^{LP/LP}$ mutants and other genotypes (Fig. 2c). Eight out of 10 mice receiving $Brca2^{LP/KO}$ fetal liver cells succumbed within 231 days of transplantation. In contrast, 9 out 10 mice receiving $Brca2^{LP/+}$ or $Brca2^{LP/LP}$ fetal liver cells and all mice receiving $Brca2^{KO/+}$ fetal liver cells remained healthy and survived for more than 8 months. To further examine the potential of donor cells supporting long-term reconstitution, we performed secondary transplantation where we harvested the bone marrow cells from the animals of the primary transplant experiment and injected them into new lethally

irradiated recipients. Here we observed that the mice receiving the $Brca2^{LP/LP}$ mutant bone marrow cells showed significantly reduced survival compared to those receiving cells from other genotypes (Fig. 2d). We were unable to perform secondary transplantation of $Brca2^{LP/KO}$ reconstituted bone marrow cells because the remaining two mice died prior to the transplantation. These findings suggest that the $Brca2^{LP/LP}$ embryonic livers have a defect in the hematopoietic stem and progenitor cells (HSPC), which are sufficient to provide reconstitution of primary recipients but not to secondary recipients. In contrast, the $Brca2^{LP/KO}$ embryonic livers are significantly deficient in HSPCs and do not support bone marrow reconstitution. These results indicate that even though the $Brca2^{L2431P}$ mutants are apparently indistinguishable from their control littermates, there is a marked difference when subjected to proliferative stress. Our findings also demonstrate that the phenotype observed in homozygous mice is less deleterious than that in hemizygous mice, which suggests that having two copies of the $Brca2^{LP}$ allele partially compensates for the defect in BRCA2-DSS1 interaction.

**$Brca2^{LP/KO}$ mice are not viable on a $Dss1$ heterozygous background.** Our previous biochemical studies have demonstrated that the leucine-to-proline change at residue 2510 of human BRCA2 significantly reduces DSS1 binding[18]. Computational studies using homology-based molecular modeling also suggested that leucine present at the core of the helical domain interacts with DSS1. To confirm whether the phenotype of $Brca2^{LP/KO}$ mice is due to reduced interaction between the BRCA2 L2431P and DSS1, we examined the effect of reducing the levels of DSS1 on the viability of $Brca2^{LP/KO}$ mice. We crossed $Brca2^{LP/KO}$ mice with $Dss1^{+/-}$ to obtain $Brca2^{KO/+};Dss1^{+/-}$ and $Brca2^{LP/+};Dss1^{+/-}$. These were subsequently intercrossed to obtain mice of various genotypes. Consistent with the impact of DSS1 on viability in other organisms[34] as well as loss of radiation-induced RAD51 foci formation in DSS1-depleted mESCs[35], we did not obtain any viable $Dss1^{-/-}$ mice (Supplementary Table 2). Surprisingly, while we obtained mice of almost all genotypes on $Dss1^{+/+}$ and $Dss1^{+/-}$ backgrounds, including $Brca2^{LP/KO};Dss1^{+/+}$ mice, we failed to obtain any viable $Brca2^{LP/KO};Dss1^{+/-}$ mice (Supplementary Table 2). This genetic evidence confirms that the phenotype of $Brca2^{LP/KO}$ mice is due to a defect in the interaction between BRCA2 and DSS1 that gets exacerbated in the $Dss1^{+/-}$ background.

**$Brca2^{LP/LP}$ and $Brca2^{LP/KO}$ mouse embryonic fibroblasts show proliferation defect and genomic instability.** To examine the

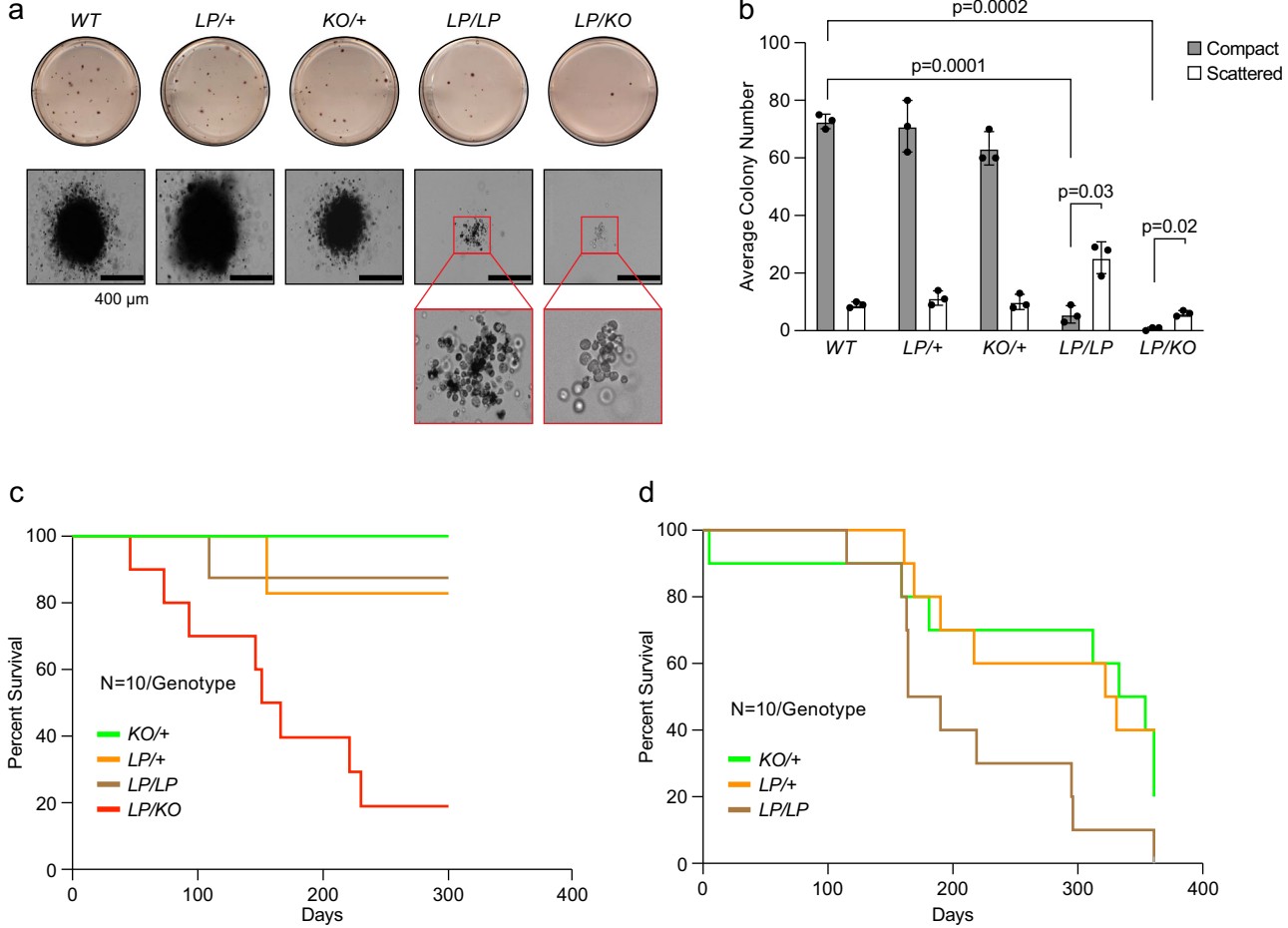

**Fig. 2 Functional characterization of hematopoietic progenitor cells from fetal liver. a** Colony-forming assay to examine proliferative potential of hematopoietic progenitors from fetal liver isolated from 16.5dpc embryos of different genotypes. *LP/LP* and *LP/KO* showed reduced number of compact colonies. Inset shows morphology of scattered colonies. **b** Quantification of colony numbers for indicated genotype shows reduced number of compact *LP/LP* and *LP/KO* colonies and higher number of scattered *LP/LP* and *LP/KO* colonies ($n = 3$ biological replicate, two-tailed Students t-Test, error bar- SD of mean). **c** Kaplan–Meier survival curve of lethally radiated recipient mice after bone marrow reconstitution by fetal liver cells of indicated genotypes. *LP/KO* fetal liver cells show reduced bone marrow reconstitution leading to shorter median survival of the recipients ($n = 10$, $p = 0.0003$, Log-rank Mantel-Cox test). All other genotypes showed no significant difference in survival. **d** Kaplan–Meier survival curve of lethally radiated recipient mice after secondary transplantation using bone marrow from primary transplantation recipients. *LP/LP* bone marrow reconstitution shows significantly reduced median survival as compared to WT ($n = 10$, $p = 0.041$, Log-rank Mantel-Cox test).

effect of disruption of the BRCA2-DSS1 interaction at the cellular level, we generated primary mouse embryonic fibroblasts (MEFs) from embryos at 13.5dpc. We found that MEFs generated from *Brca2^{LP/LP}* and *Brca2^{LP/KO}* embryos grew significantly slower than the control MEFs (Fig. 3a). Cell cycle analysis showed an increase in cell stalling at the G2/M phase in the mutant MEFs, suggesting the persistence of unrepaired DSBs results in slower growth rate (Fig. 3b). We examined the genomic integrity of MEFs and did not observe any significant difference between any of the genotypes (Fig. 3c, d). However, when we challenged the MEFs with 100 nM MMC (DNA crosslinking agent), both *Brca2^{LP/LP}* and *Brca2^{LP/KO}* MEFs exhibited a significant increase in the number of chromosomal aberrations (Fig. 3c, d). Notably, the number of radials (characteristic chromosomal defect associated with FA[36]) observed in mutant MEFs were significantly increased after MMC treatment (Supplementary Fig. 3a). This suggests that despite the observed slower growth, the mutant MEFs were able to repair DNA damage during the normal cell cycle. However, when challenged with a DNA-damaging agent, the mutant MEFs exhibited significant deficiency in their ability to maintain genomic integrity.

**Brca2^{LP/LP} and Brca2^{LP/KO} MEFs are deficient in radiation-induced RAD51 foci formation.** We have previously reported a marked reduction in ionizing radiation (IR)-induced RAD51 foci formation and HR in mESCs expressing the BRCA2 L2510P variant[18]. We exposed MEFs of all genotypes to 10 Gy of IR and examined RAD51 foci formation at 1, 3, 5, and 7 h post-IR. We examined γH2AX foci formation as a pan-DSB marker and quantified the percentage of cells with at least five RAD51 foci that co-localized with γH2AX foci. We observed an increase in the percentage of RAD51 foci positive cells in *Brca2^{+/+}*, *Brca2^{KO/+}*, and *Brca2^{LP/+}* MEFs at 1 h (40–50%) and 3 h (60–70%), followed by a gradual reduction at 5 h (50–55%) and 7 h (30–40%) (Fig. 4a, b). In contrast, we observed 3–7% of cells at 1 h, 5–10% at 3 h, 2–4% at 5 h, and 1–2% at 7 h in *Brca2^{LP/LP}* and *Brca2^{LP/KO}* MEFs. These results demonstrate a significant reduction in RAD51 recruitment at IR-induced DSBs in *Brca2^{LP/LP}* and *Brca2^{LP/KO}* MEFs.

The interaction between BRCA2 and DSS1 is essential for the removal of RPA32 from 3' ssDNA overhangs at DSBs. This is crucial for RAD51 loading and repair by HR. Therefore, we examined the recruitment and removal of RPA32 in response to

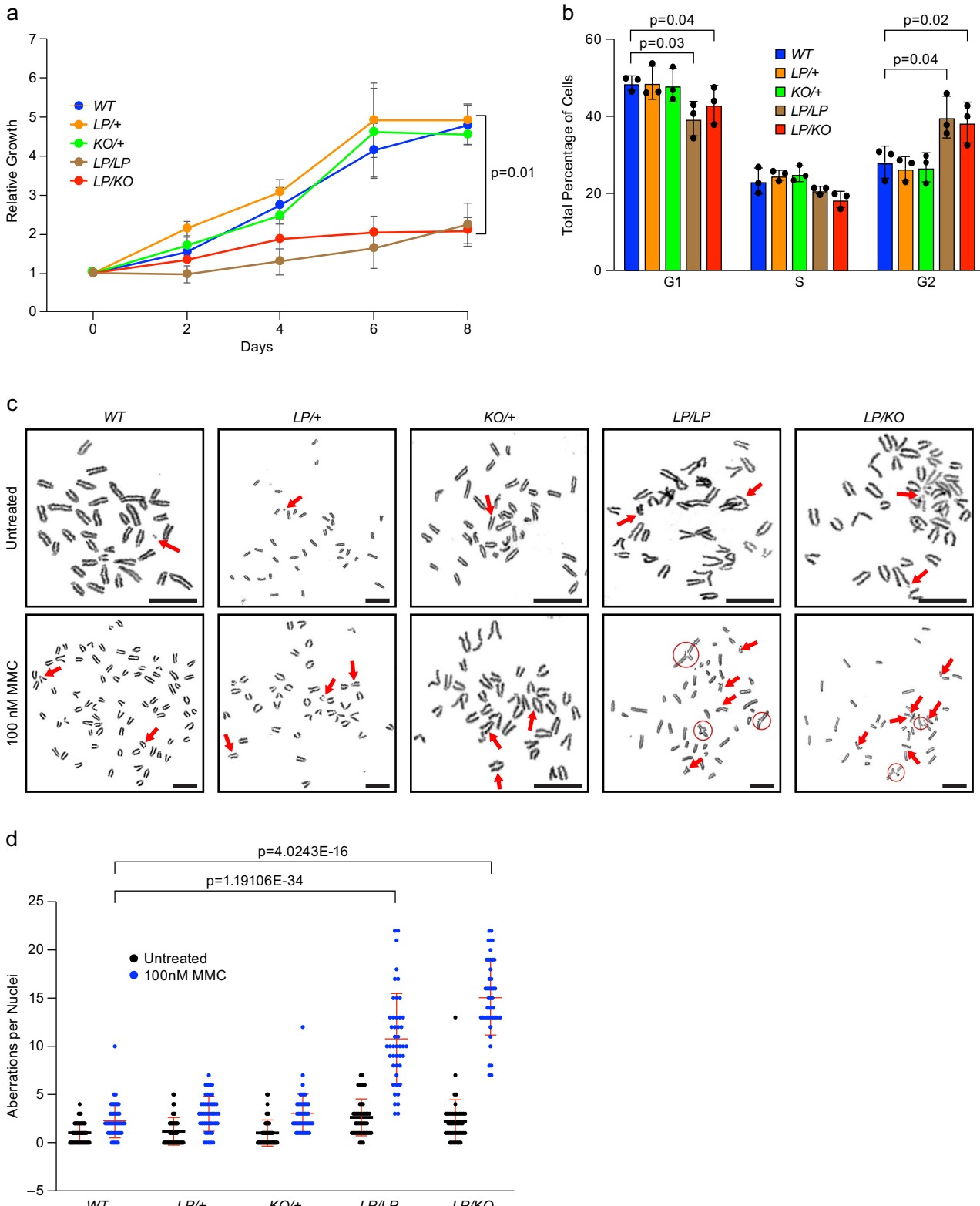

**Fig. 3 *Brca2^{L2431P}* mutant MEFs exhibit genomic instability. a** Growth curve of MEFs of different genotypes over 8 days. *LP/LP* and *LP/KO* MEFs have reduced relative growth compared to controls ($n = 3$ biological replicate, one-tailed t-test: two-sample assuming unequal variances, error bar- SE of mean). **b** Cell cycle analyses of MEFs of indicated genotypes shows *LP/LP* and *LP/KO* MEFs get stalled in G2 phase ($n = 3$ biological replicate, two-tailed Student's *t* test, error bar- SD of mean). **c** Chromosomal aberrations in MEFs of indicated genotype, untreated (top panels) and after 100 nM MMC treatment (lower panels). Aberrations are marked with red arrows and radials are circled (scale bar = 5 μm). **d** Quantification of chromosomal aberrations in MEFs shown in **c**. *LP/LP* and *LP/KO* MEFs exhibit increased number of chromosomal aberrations after treatment as compared to WT ($n = 3$ individual MEFs with 15 nuclei each making 45 nuclei in total, two-tailed Student's *t* test, error bar- SD of mean).

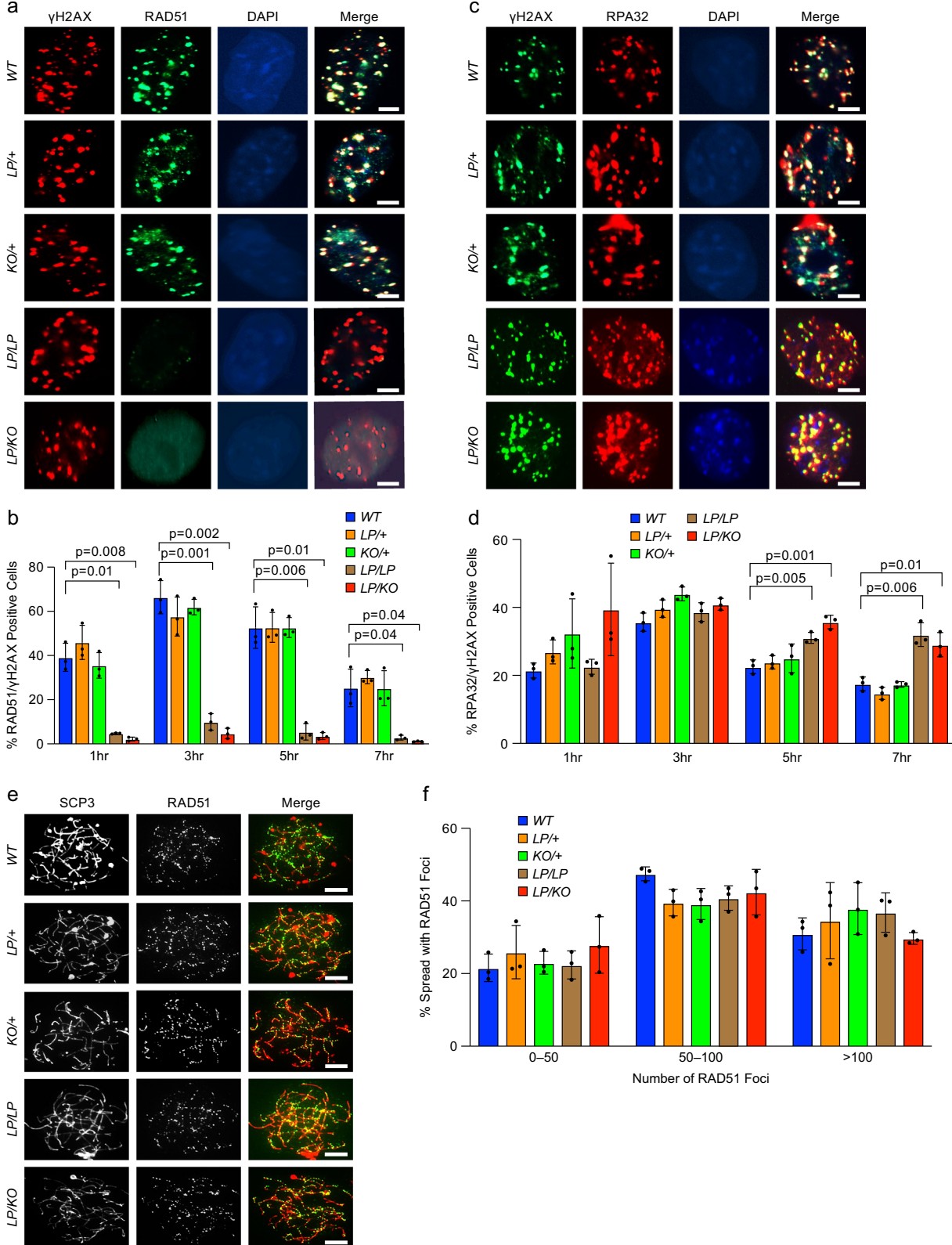

IR in the MEFs. We observed an increase in the percentage of cells showing RPA32 foci in all genotypes at 1 h (20–35%) and 3 h (35–45%). Interestingly, while in the *Brca2*<sup>+/+</sup>, *Brca2*<sup>KO/+</sup>, and *Brca2*<sup>LP/+</sup> MEFs we observed a reduction in RPA32 positive cells at 5 and 7 h post-IR, there was a significant persistence of RPA32 foci in *Brca2*<sup>LP/LP</sup> and *Brca2*<sup>LP/KO</sup> MEFs. These findings suggest a

defect in RPA32 removal in cells expressing the BRCA2 L2431P variant, which is likely to impede RAD51 recruitment (Fig. 4c, d).

Another major function of BRCA2 is protection of the stalled replication forks from degradation by nucleases such as MRE11 and EXO1[7,37]. To examine the impact of the BRCA2 L2431P variant on protection of stalled forks, we performed DNA fiber

**Fig. 4 *Brca2^L2431P* mutant MEFs are deficient in RAD51 loading at radiation-induced DSBs. a** Representative immunofluorescence images of MEFs of indicated genotypes showing RAD51 foci after 3 h of 10 Gy IR. DSBs are marked with γH2AX and nuclei with DAPI. *LP/LP* and *LP/KO* MEFs are deficient in RAD51 foci (scale bar = 5 μm). **b** Quantification of RAD51-positive nuclei per γH2AX positive nuclei at different time points (n = 3 biological replicate, two-tailed Student's *t* test, error bar- SD of mean). **c** Representative immunofluorescence images of MEFs of indicated genotypes showing RPA32 foci after 3 h of 10 Gy IR. DSBs are marked with γH2AX, and nuclei are stained with DAPI. MEFs show visible RPA32 foci (scale bar = 5 μm). **d** Quantification of RPA32 positive nuclei per γH2AX positive nuclei at different time points (n = 3 biological replicate, two-tailed Student's *t* test, error bar- SD of mean). **e** Representative immunofluorescence images of RAD51 foci in spermatocyte spreads from testes of 4-week-old mice of indicated genotypes in leptotene/zygotene stages. The synapsed chromosomes are labeled with SCP3 (scale bar = 5 μm). **f** Quantification of RAD51 foci observed per spermatocyte (n = 3 biological replicate, error bar- SD of mean).

analysis using the MEFs of all genotypes, with *Brca1del11* (*Brca1^Δ11/Δ11*) MEF as a control for unprotected replication forks[38,39]. We pulse-labeled the MEFs with thymidine analogs, first with IdU (red labeled) then by CldU (green labeled), followed by treatment with hydroxyurea (HU). A reduction in the CldU track lengths relative to that of IdU suggests loss of fork protection. We found the ratio of CldU to IdU track lengths close to 1 for MEFs of all genotypes except *Brca1del11*, suggesting that *Brca2^LP/LP* as well as *Brca2^LP/KO* MEFs are proficient in protecting stalled replication forks (Supplementary Fig. 3b).

**Brca2^LP/KO mice have normal meiotic progression and RAD51 recruitment in spermatocytes.** Given the fact that the BRCA2 L2431P variant has a significant impact on RAD51 recruitment to the DSBs, it is puzzling that *Brca2^LP/LP* and *Brca2^LP/KO* mice are fertile. Previous studies have shown that a defect in BRCA2-mediated DSB repair results in male infertility[40,41]; testes of mutant mice are reduced in size and seminiferous tubules show a significant depletion of the germ cells[40,42]. We did not observe any difference in the size of the testes of adult males of various genotypes, including *Brca2^LP/LP* and *Brca2^LP/KO*. Histological analysis of the testes confirmed normal morphology and presence of sperm in the seminiferous tubules (Supplementary Fig 4a, upper panel). TUNEL staining also did not reveal any increase in the number of apoptotic cells (Supplementary Fig 4a, lower panel). Similarly, we did not observe any difference in the H&E-stained sections of ovaries of different genotypes (Supplementary Fig 4b).

We examined the impact of the L2431P variant on meiotic progression. During the prophase I of meiosis, SPO11 generates DSBs that are repaired by HR, similar to the repair of DSBs in somatic cells by HR[43]. We examined spermatocyte spreads from all genotypes at leptotene stage, when the DSBs are generated and RAD51 foci formation starts to appear. We detected the DSBs by γH2AX staining and visualized the paired chromosomes with SCP3, a marker of the synaptonemal complex. Surprisingly, spermatocyte spreads from *Brca2^LP/LP* as well as *Brca2^LP/KO* mice exhibited a normal number of RAD51 foci with no significant difference compared to the controls (an average of 50–150 RAD51 foci per spermatocyte, Fig. 4e, f). Furthermore, we observed similar number of persistent γH2AX foci in the late pachytene stage in all genotypes, suggesting that the DSBs have been repaired and meiotic progression was normal (Supplementary Fig 4c, d). At this stage, the presence of γH2AX cloud is visible only on the unsynapsed sex chromosomes. Overall, we did not detect any defect in the meiotic progression in *Brca2^LP/LP* and *Brca2^LP/KO* mice. These observations suggest that decrease in BRCA2-DSS1 interaction does not affect RAD51 recruitment to DSBs during meiosis and the meiotic HR is unhampered.

**Mutant MEFs show RAD51 foci at replication-induced DSBs: somatic vs. meiotic homologous recombination.** The clear difference in the RAD51 recruitment between the MEFs and spermatocytes is intriguing. On one hand, BRCA2 L2431P mutant

MEFs are severely deficient in RAD51 recruitment after IR-induced DNA damage. On the other hand, RAD51 recruitment during meiosis is unaffected with this mutation. By over-expressing *DSS1* cDNA in the MEFs (Supplementary Fig. 5a), we ruled out the possibility that higher expression of DSS1 in the spermatocytes may contribute to this difference. Increased expression of DSS1 did not restore the IR-induced RAD51 foci formation defect in *Brca2^LP/LP* and *Brca2^LP/KO* MEFs (Supplementary Fig. 5b, c). Conversely, *Dss1* heterozygosity did not have any effect on RAD51 loading in *Brca2^LP/LP* spermatocytes (Supplementary Fig. 5d, e).

The key difference between repair of Spo11-generated DSBs during meiosis and the repair of IR-induced DSBs by HR is the proximity of the homologous DNA to the DSBs. During meiosis, spatial alignment of homologous chromosomes occurs at the pre-leptotene stage prior to the generation of DSBs. It is restricted to the telomeric region of the homologous chromosomes and is referred to as the bouquet stage[44,45]. Although the homologous chromosomes are not fully paired, they are present in close proximity, which may aid in the homology search during DSB repair[46]. In contrast, during the repair of IR-induced DSBs by HR, the homologous DNA is not present in close proximity. Instead, the RAD51-coated nucleofilaments search for the homologous DNA and then displace the complementary strands to initiate recombination[5] (Supplementary Fig. 6a, b).

Based on this difference, we hypothesized that the presence of a homologous DNA in close proximity of DSBs may circumvent the requirement for DSS1-bound BRCA2. It has been reported that low concentration of camptothecin (25–30 nM) generates single-strand breaks (SSB)[47]. These SSBs are converted into DSBs during DNA replication[48], and the sister chromatid is present close to the replication-induced DSBs (Fig. 5a). This scenario may resemble the meiotic DSB repair by HR, where the homologous DNA is in close proximity to the DSBs. We checked the DSB formation in WT MEFs by treating them with 10 Gy IR, 30 nM camptothecin, and 1 μM camptothecin for 1 h, along with 1 mM EdU (to label replicating DNA). We found more than 95% of DSBs generated in low-dose camptothecin (30 nM) treatment were S-phase specific, in contrast to 58% in cells treated with high-dose camptothecin (1 μM) and 35% in cells treated with IR (Fig. 5b and Supplementary Fig. 7a). These DSBs were also confirmed by the presence of 53BP1 foci after 30 nM camptothecin treatment (Supplementary Fig. 7b, c). Surprisingly, 30 nM camptothecin treatment for 24 h resulted in RAD51 foci formation in *Brca2^LP/LP* as well as *Brca2^LP/KO* MEFs (Fig. 5c). Although the percentage of RAD51-positive nuclei (30–35%), was lower compared to the controls (45–55%), there was a significant increase in percentage of RAD51-positive nuclei over what was observed after IR (Fig. 5d). To further validate this observation, we generated AFs from ear punch biopsies. Interestingly, we observed a significant increase in RAD51-foci-positive nuclei in both *Brca2^LP/LP* (50% vs. 22%) and *Brca2^LP/KO* (34% vs. 7%) fibroblasts after camptothecin treatment as compared to IR (Supplementary Fig. 8).

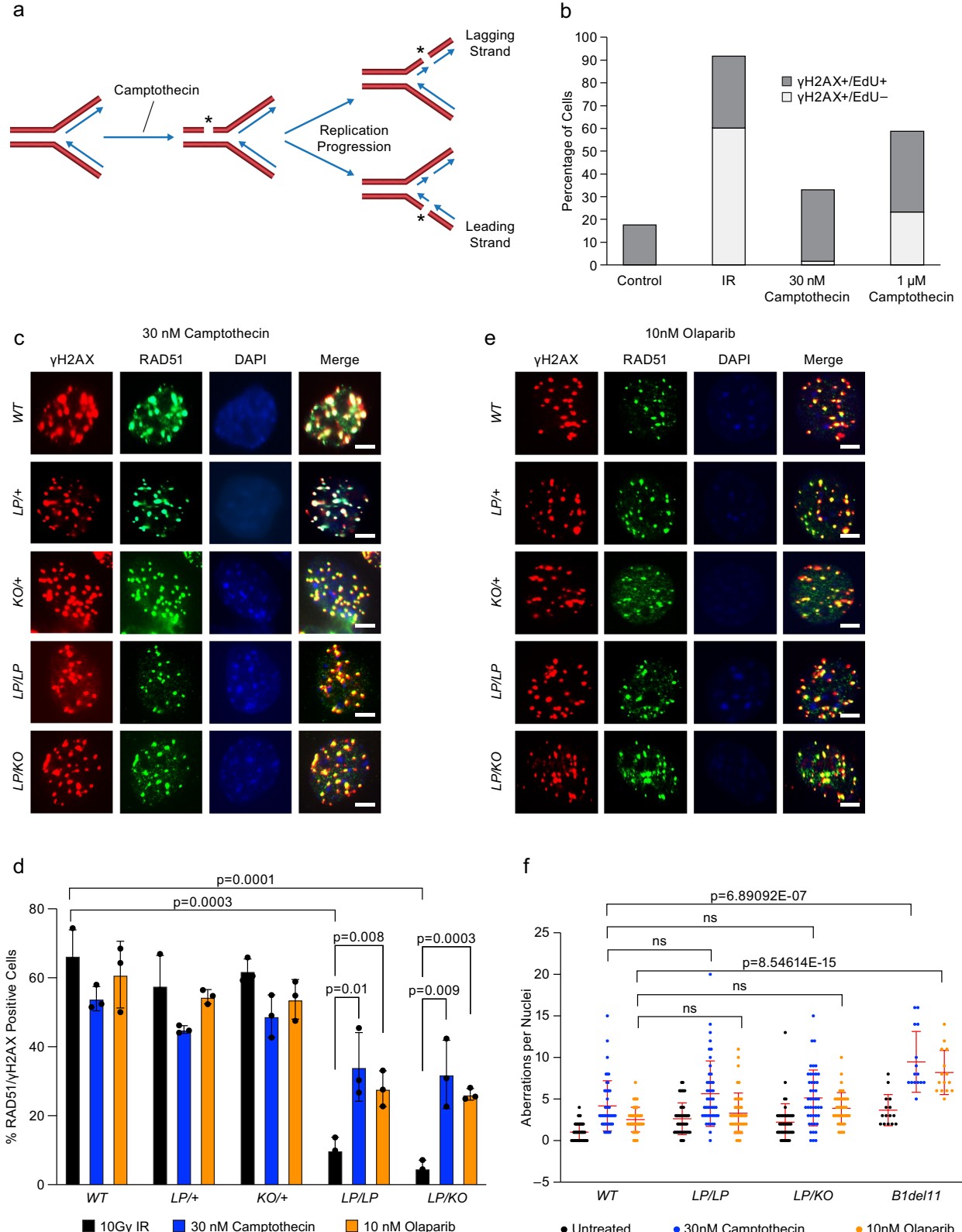

PARP proteins are essential for the repair of DNA single strand breaks[49,50]. PARP inhibitors, such as olaparib, inhibit repair of single-strand breaks and result in accumulation of replication-induced DSBs[51]. Therefore, we used olaparib to test our hypothesis that RAD51 is efficiently recruited at the replication-induced DSBs. MEFs of all the genotypes were treated with 10 nM olaparib for 24 h and examined for RAD51 foci formation. As expected, we observed

a significant increase in RAD51-foci-positive *Brca2*[LP/LP] and *Brca2*[LP/KO] MEFs compared to irradiated MEFs (Fig. 5d, e). Taken together, these results suggest that there is a difference in the requirement of BRCA2-DSS1 interaction between IR and camptothecin or olaparib-induced DSB repair via HR. It is possible that when the homologous DNA is present near the DSB, BRCA2-DSS1 interaction is dispensable for RAD51 loading at DSBs.

**Fig. 5 *Brca2^L2431P* mutant MEFs are proficient in RAD51 loading at low-dose camptothecin-induced DSBs. a** Schematic representation of how an unrepaired single strand break generated by Topoisomerase inhibitor (camptothecin) gets converted into a DSB during DNA replication. The undamaged sister chromatid is present near the DSB, which acts a template for HR-mediated DSB repair. **b** quantification of γH2AX positive nuclei in MEFs co-stained with EdU after 10 Gy IR, 30 nM camptothecin, and 1 μM camptothecin treatment for 1 h grown in 1 mM EdU media. Notably, more than 95% of DSBs seen after 30 nM camptothecin treatment is in S-phase (similar to control) as opposed to other treatments. **c** Representative immunofluorescence images of RAD51 foci in MEFs of indicated genotypes after 30 nM camptothecin treatment for 24 h. DSBs are marked with γH2AX, and nuclei are stained with DAPI (scale bar = 5 μm). **d** Quantification of RAD51-positive nuclei per γH2AX positive nuclei in MEFs after different treatments. RAD51 loading after IR is significantly reduced in *LP/LP* and *LP/KO* MEFs compared to WT (*n* = 3 biological replicate, two-tailed Student's *t* test, error bar- SD of mean). Mutant MEFs show increased number of RAD51-positive nuclei after camptothecin and olaparib treatment (error bar- SD of mean). **e** Representative immunofluorescence images of RAD51 foci in MEFs of indicated genotypes after 10 nM olaparib treatment for 24 h. DSBs are marked with γH2AX, and nuclei are stained with DAPI (scale bar = 5 μm). **f** Quantification of chromosomal aberration observed in untreated, 30 nM camptothecin and 10 nM olaparib treated MEFs of indicated genotypes (*n* = 3 individual MEFs with 15 nuclei each making 45 nuclei in total, ns not significant, two-tailed Student's *t* test, error bar- SD of mean).

To test whether the RAD51 foci induced by camptothecin or olaparib are functional and efficient in repairing the damaged DNA, we examined their impact on the genomic integrity of *Brca2^LP/LP* and *Brca2^LP/KO* MEFs. These treatments resulted in an increase in number of chromosomal aberrations in HR-deficient *Brca1* mutant MEF (*Brca1del11*)[39] (Fig. 5f). However, similar to the *Brca2^+/+* control MEFs, *Brca2^LP/LP* and *Brca2^LP/KO* MEFs did not exhibit any significant increase in the number of chromosomal aberrations (Fig. 5f and Supplementary Fig 7d). These results strongly suggested that the DNA damage induced by camptothecin and olaparib during DNA replication is efficiently repaired by the BRCA2 L2431P variant and does not increase the genomic instability of mutant cells.

**Donor template in the vicinity of DSB partially rescues HR defect.** To further test our hypothesis that the presence of a homologous DNA in the vicinity of DSB can restore HR in cells expressing the BRCA2 L2510P variant, we developed a CRISPR-Cas9-based assay to study HR by reconstituting a mutant blasticidin resistance cassette. mESCs that are sensitive to blasticidin, owing to a 29 bp deletion (Fig. 6a and Supplementary Fig. 9a), become resistant if they are able to perform HR and restore the deleted region using a double-stranded donor template. Streptavidin-fused Cas9 was used to tether biotinylated donor template (cloned in circular plasmid) to mimic the availability of the donor DNA in proximity to the DSB site (Fig. 6b). We generated a *Brca2^KO/KO* mESC line (with 29 bp deletion in the blasticidin resistance gene) expressing the human *BRCA2^L2510P* variant cloned in a BAC, and co-transfected the cells with Cas9-mSA[52] (tagged with monomeric streptavidin) and an sgRNA specific to the blasticidin resistance gene to induce a DSB (Fig. 6b). When a biotinylated homologous donor DNA was provided to repair the deleted region, blasticidin-resistant colonies increased by 1.6-fold compared to a non-biotinylated donor (Fig. 6c, d) in cells expressing WT BRCA2. This suggests that the streptavidin-biotin interaction brings the donor template in the vicinity, thereby increasing the HR efficiency, as also previously reported in other cell lines[53,54]. *Brca2^KO/KO* mESCs expressing the *BRCA2^L2510P* variant yielded a significantly lower number of blasticidin-resistant colonies when provided with a non-biotinylated donor template (Fig. 6c, d and Supplementary Fig 9b). However, in the presence of biotinylated donor DNA, we observed a 2.67-fold increase in the number of blasticidin-resistant colonies. These results suggest that HR gets partially restored in *BRCA2^L2510P* cells when the donor DNA, tethered by Cas9 via a streptavidin-biotin interaction, is present in the vicinity of the DSBs.

**Discussion**
We have previously reported *BRCA2^L2510P*, an FA-associated variant, to be pathogenic, deficient in HR and sensitive to DNA-damaging drugs. This is because the *BRCA2^L2510P* variant disrupts the interaction of DSS1 with BRCA2[18]. In the present study, we have generated a *Brca2* knock-in mouse model to examine ways in which the disruption of BRCA2-DSS1 interaction impacts growth and development, tumorigenesis, and other biological processes. We observed embryonic lethality in a significant fraction of homozygous and hemizygous mutant mice. However, several homozygous and hemizygous mice were also born alive. On a *Dss1^+/−* background, we failed to obtain any viable *Brca2^LP/KO* mice, which provides genetic evidence that the lethality among these mice is due to the disruption of interaction with DSS1.

Despite the significant reduction in the viability of *Brca2^LP/KO* and *Brca2^LP/LP* mice, those that are born alive grew normally and did not show increased tumor susceptibility. However, *Brca2^LP/KO* mice have lower body weights, shorter life span, and hypersensitivity to IR. The hemizygous mutants' radiation sensitivity suggests that they have severely reduced potential for bone marrow progenitor renewal, as sublethal IR rapidly depletes bone marrow cells and the WT mice are able to recover and survive[55]. This was evident when we performed a fetal liver hematopoietic cell colony-forming assay and bone marrow transplant of fetal liver cells. In both these experiments, the hemizygous mutants exhibited severe defects and the homozygous *Brca2^LP/LP* animals exhibited milder phenotypes, suggesting that having an additional copy of the variant can partially complement some of the defects observed in hemizygous mice.

Functional evaluation of the impact of the L2431P variant in MEFs confirmed a significant reduction in IR-induced RAD51 foci formation, which is a clear indicator of HR defect. Displacement of RPA32 nucleofilaments, with the help of DSS1 bound to the BRCA2 helical domain, is crucial for RAD51 loading and foci formation[56], which makes this BRCA2-DSS1 interaction indispensable for RAD51 loading and eventual HR[57]. Consistent with this, we observed delayed removal of RPA32 from the damaged sites after radiation in the mutant MEFs. Furthermore, there was no rescue of the RAD51 loading defect in mutant MEFs even after DSS1 overexpression. These results demonstrated that the L2431P variant has a severe HR defect.

Based on the severe HR defect, it is surprising that *Brca2^LP/LP* and *Brca2^LP/KO* mice are fully fertile and had no observable defect in their gonads. It is well established that BRCA2-mediated HR is required for normal meiotic progression across species[58–61]. BRCA2 repairs Spo11-generated DSBs by HR during meiotic prophase I, similar to the HR-mediated repair in somatic cells[58]. We have previously reported that defects in DNA repair machinery due to BRCA2 deficiency[40] or mutation[42] leads to infertility and azoospermia. Our study with meiotic spreads of spermatocytes revealed no HR defect in *Brca2^LP/LP* and *Brca2^LP/KO* mice. The progression of HR during meiosis I is very similar to that in somatic cells, as both processes converge on a common mechanism and utilize similar factors for their progression. The absence of

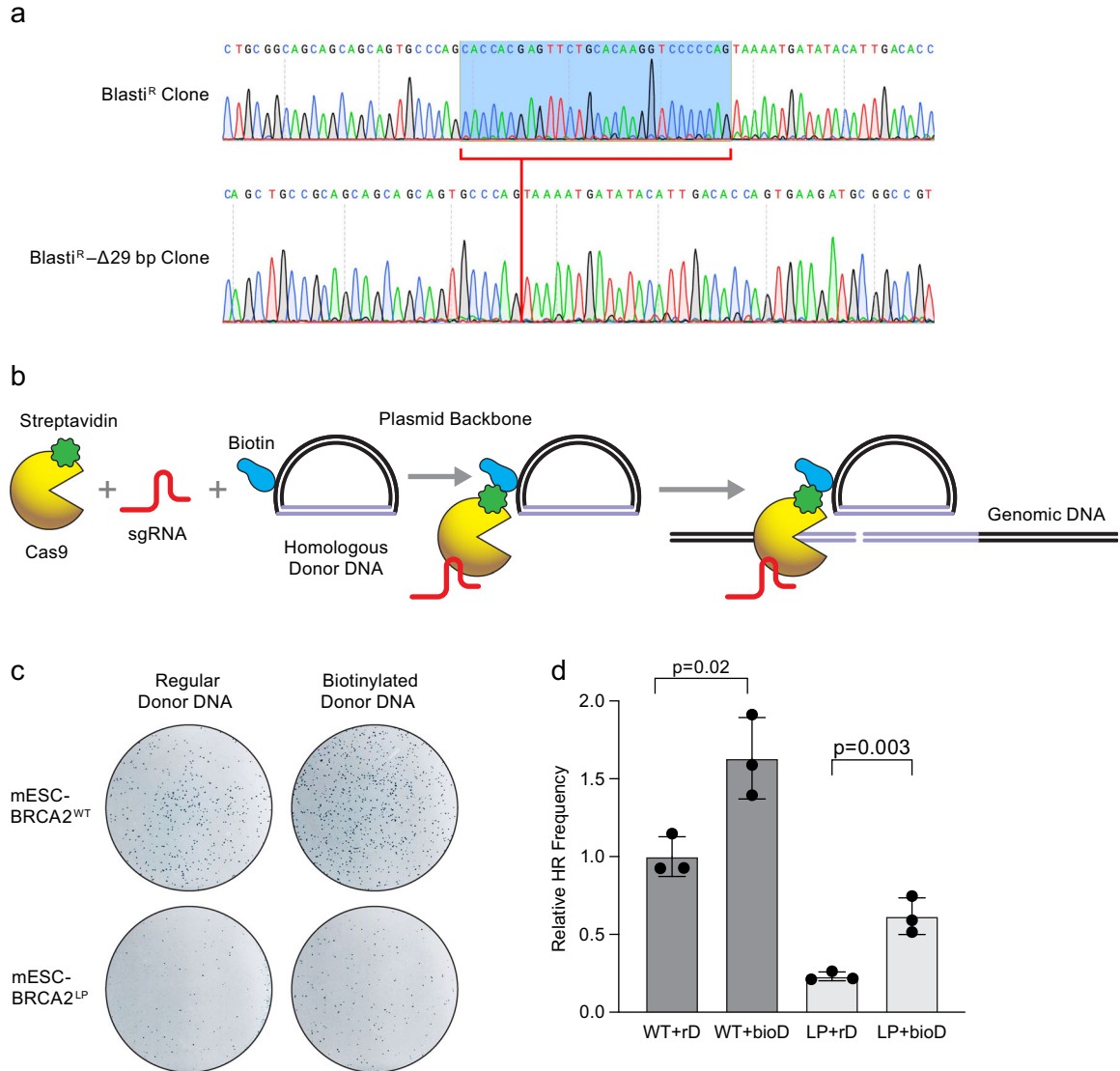

**Fig. 6 CRISPR Cas9-based assay to determine HR efficiency. a** Chromatograms (reverse complementary strand) showing full-length and 29 bp deletion in the blasticidin resistance gene which renders the cells sensitive to blasticidin. **b** Schematic representation of streptavidin-tagged Cas9-mediated delivery of biotinylated donor DNA near to the DSB (generated by Cas9) to assist HR. **c** Representative plates showing mESC colonies obtained in WT and BRCA2[L2510P] complemented *Brca2*[KO/KO] mESCs after HR-mediated repair of mutant blasticidin-resistance gene provided with regular or biotinylated donor DNA. **d** Quantification of relative HR efficiency in BRCA2[WT] and BRCA2[L2510P] expressing mESCs. Number of blasticidin-resistant colonies are normalized to colonies in plates without selection, rD: regular donor, bioD: biotinylated donor (see Supplementary Fig. 9b, $n = 3$ technical replicate, two-tailed Student's *t* test, error bar- SD of mean).

functional BRCA2 or RAD51 impairs HR in both somatic and germ cells[40,42,62]. During meiosis, the homologous chromosomes are present in close proximity prior to the repair of DSBs. In contrast, in somatic cells, homology search and pairing is done by RAD51 nucleofilaments[1,2,4,43]. Moreover, HR in somatic cells occurs largely with sister chromatids generated as a result of DNA replication during the S-phase[63], whereas HR in meiosis I takes place between homologous chromosomes and rarely with the sister chromatids.

This dichotomy in the HR mechanism in somatic cells and meiosis prompted us to simulate meiosis-like conditions in somatic cells. To simulate this, we used a low dose of camptothecin and olaparib treatment to induce DNA SSBs; during DNA replication, these get converted into DSBs at the replication fork when the homologous DNA (sister chromatid) is still nearby[64,65]. Interestingly, we observed distinct RAD51 foci in mutant MEFs treated with camptothecin and olaparib. The rescue

of the RAD51 loading defect was also observed in mutant AFs with similar treatment. While HR-defective *Brca1* mutant MEFs exhibited marked increase in genomic instability in response to camptothecin and olaparib, *Brca2*[LP] mutant MEFs were able to maintain their genomic integrity.

The HR assay using streptavidin-fused Cas9 to repair the blasticidin resistance gene strengthens our hypothesis that the presence of homologous DNA in the vicinity of DSB can bypass the need for BRCA2-DSS1 interaction for efficient RPA32 removal, RAD51 loading, and eventual HR. Our findings are consistent with the previously reported mechanism of RPA32 removal from ssDNA, where DSS1 acts as a DNA mimic, via its acidic domain, to attenuate the affinity of RPA32 towards ssDNA, leading to its removal from ssDNA[57]. Mutation in the acidic domain of DSS1 resulted in defective RPA32 unloading[57]. Thus, if actual homologous DNA (instead of a mimic) is nearby, as in case of replicating DNA (assessed by camptothecin and olaparib

treatment) or biotinylated donor DNA attached to streptavidin-fused Cas9, RPA32 removal is no longer dependent on DSS1 bound to BRCA2. This mechanism is responsible for successful HR-mediated DNA repair during normal animal growth and genomic stability.

We propose that the cells expressing the L2510P/L2431P variant are proficient in HR-mediated repair of DSBs generated during normal DNA replication. However, under conditions of rapid proliferative stress (such as during embryogenesis or bone marrow transplantation) or in response to DNA-damaging agents (such as IR and MMC), cells are unable to repair DSBs by HR efficiently. As a result, most $Brca2^{LP/LP}$ and $Brca2^{LP/KO}$ mice die during embryogenesis. The mice that are born alive develop normally and exhibit no overt phenotypes. They do not accumulate significant DNA damage because the interaction between DSS1 and BRCA2 is dispensable for DSB repair during the S-phase. Therefore, these mice are not predisposed to tumorigenesis. Our hypothesis is supported by two reports suggesting that BRCA2 W2626C, which also disrupts the interaction between BRCA2 and DSS1, is a low-risk variant[66,67]. We and others have shown that, similar to L2510P, the W2626C variant is defective in RAD51 recruitment and has a significant impact on cell viability and HR[18]. Future studies will reveal the cancer risk in individuals carrying BRCA2 L2510P and other variants that disrupt this interaction and are considered HR-deficient.

## Methods

**Generation of $Brca2^{L2431P}$ knock-in mice.** $Brca2^{L2431P}$ knock-in mice were generated by gene targeting approach in mouse V6.4 ES cells. Gene targeting construct was generated by recombineering based approach as described previously[68]. Briefly, a bacterial artificial chromosome containing full-length $Brca2$ (RPCI22-421-23A) was used to mutate two nucleotides of exon 15 of $Brca2$ to change codon 2431 from CTG to CCA, (leucine to proline) using a $galK$-based selection-counterselection method[69,70]. Next, a $loxP$-PGK-Tn5-$neomycin$-bp(A)-$loxP$ cassette along with $Spe$I and $Eco$RV restriction sites were inserted into intron16 by recombineering. Finally, a 13.5 kb genomic region of $Brca2$ was retrieved by gap repair from the BAC into pBSK(+) plasmid containing the $Thymidine Kinase$ (TK) gene under the control of the MC1 promoter. The knock-in targeting vector was linearized with $Not$I and electroporated into V6.4 ES cell line as described previously. G418 and FIAU resistant ES cell clones were screened by Southern analysis of $Spe$I and $Eco$RV restriction enzyme digested genomic DNA using 5′ and 3′ probes (Supplementary Fig. 1). Some correctly targeted ES cell clones were injected into C57BL/6 blastocysts to generate chimeras. One of these chimeras transmitted the targeted allele in the germ line and $Brca2L^{2431P-Neo}/+$ pups were obtained.

**Generation of $Brca2^{L2431P}$ mice of various genotypes.** To obtain $Brca2^{L2431P/+}$ mice ($Brca2^{LP/+}$), $Brca2^{L2431P-Neo}/+$ mice were crossed to β-actin-Cre-deleted mice[71]. $Brca2^{LP/+}$ mice were intercrossed to generate $Brca2^{LP/LP}$ mice. To obtain $Brca2^{LP}$ mice on a pure background, $Brca2^{LP/+}$ mice on mixed C57BL/6 and 129/SvEv background were backcrossed to C57BL/6 mice for 10 generations. To generate $Brca2^{LP}$ hemizygous mice ($Brca2^{LP/KO}$), $Brca2^{LP/+}$ mice were crossed with $Brca2^{KO/+}$ mice carrying the null allele ($KO$)[33].

$Dss1^{KO/+}$ mice were generated using cryopreserved sperm from $Dss1^{KO/+}$ ($Sem1^{KO/+}$ obtained from RIKEN, ID #RBRC10583) by in vitro fertilization. $Dss1^{KO/+}$ mice were crossed with $Brca2^{KO/+}$, $Brca2^{LP/+}$ to obtain mice of various genotypes used in the study. These mice we maintained on C57BL/6 pure genetic background. All animal studies were performed as per the protocols outlined in the Guide for the Care and Use of Laboratory Animals and approved by the NCI-Frederick Animal Care and Use Committee. All animal studies were performed in compliance with ARRIVE (Animal Research: Reporting In Vivo Experiments) guidelines (https://arriveguidelines.org/arrive-guidelines).

**Ethics statement.** All mice were housed, bred and used in the study following the recommendations of the Guide for the Care and Use of Laboratory Animals (The National Academies Press; 8th edition). The study protocol was approved by the Animal Care and Usage Committee (ACUC) of NCI-Frederick (Animal Study# 18–471). Animals were maintained at a 12-h light/dark cycle. Rooms were maintained at temperatures in the range of 20–27 °C and 30–70% relative-humidity.

**Genotyping.** For genotyping, genomic DNA was extracted from tail biopsies by lysing the cells using Proteinase K at 55 °C following standard procedures[33]. Genotyping was performed by PCR using 50 ng of DNA with the Taq DNA polymerase using primers listed in Supplementary Table 3.

**Animal survival study post IR.** Six to eight weeks old animals from each genotype were given 8 Gy of γ-irradiation using a Cs source. Following irradiation, mice were closely monitored. Mice showing signs of distress including weight loss were euthanized following ACUC guidelines.

**Fetal liver cells colony-forming assay.** Embryos at 16.5dpc were collected and euthanized. Livers of these embryos were carefully excised and tweezed in plain IMDM. The cell suspension was strained through a 40 μm cell strainer. 25,000 liver cells were plated in 1 ml MethoCult (M3231, stem cell technologies) media with growth factors (10% FBS, 100 ng/ml muSCF,100 ng/ml huTpo, 100 ng/ml huFlt3L, 50 ng/ml muIL6, 30 ng/ml muIL3, all from PeproTech). These were allowed to grow to form visible colonies (7–10 days). Colonies were stained with Iodonitrotetrazolium chloride (sigma I10406).

**Fetal liver cell transplantations.** Fetal liver (FL) cells were harvested from the E16.5 embryos and dissociated at 4 °C in PBS by mechanical disruption and femurs were flushed using a 5 ml syringe with a 29 G × ½ needle. The cells were passed through 45 μm mesh filters (Millipore) and resuspended in PBS at $1 \times 10^6$ cells/0.2 ml. The FL and bone marrow (BM) cell suspension were injected into the tail vein of lethally irradiated (10 Gy using a Cs source) primary and secondary recipient mice (B6.SJL-Ptprca Pep3b/BoyJ, CD45.1, Charles River Laboratories).

**Carmine Alum staining of mammary gland.** Mammary glands were harvested from 4–6-weeks-old female mice and fixed in Carnoy's solution (6:3:1 of 100% ethanol, chloroform, and glacial acetic acid). The glands were rehydrated with decreasing grades of ethanol wash and stained with Carmine Alum (Sigma C1022) overnight. The glands were subsequently dehydrated in increasing grades of Ethanol and incubated in Xylene substitute for tissue clearing for 2–3 days. The whole-mount glands were imaged using a brightfield microscope (Zeiss Axiocam) and analyzed by Carl Zeiss zenblue2.6.

**Generation of mouse embryonic fibroblasts.** E13.5 embryos from timed mating of specific genotypes were obtained. These embryos were genotyped and finely chopped in 0.5% trypsin EDTA solution (15400-054, Gibco) and incubated for 30 min at 37 °C. Digested slurry was plated in 100 mm culture plates in DMEM + 10% FBS. Once confluent these P0 (passage zero) cells were frozen in small aliquots in liquid nitrogen.

**Generation of adult fibroblasts.** In all, 5–10 mm ear punch biopsies of the animals of desired genotype were obtained. These were washed briefly in 70% ethanol and rinsed three times with sterile Hank's balanced salt solution (HBSS). These ear punches were chopped finely in collagenase (C7657, sigma) solution (2000 IU/ml dissolved in HBSS) and incubated at 37 °C for 3 h. Digested slurry was centrifuged to remove the supernatant. The pellet was resuspended and incubated in 0.5% trypsin EDTA solution (15400-054, Gibco) for 30 min at 37 °C. Digested ear punches are plated in 100 mm culture plates in DMEM + 10% FBS. Once confluent these P0 cells were frozen in small aliquots in liquid nitrogen.

**Cell proliferation and cell cycle analyses.** In total, $10^5$ cells (MEFs) of each genotype were plated in a six-well plate in triplicate and viable cell counting using trypan blue was done at the indicated time points. MEFs of all genotypes were treated with Fxcycle PI/RNase solution (Invitrogen, F10797) as per manufacturer's guidelines and passed through BD FACS (BD LSRII). The results were analyzed using FlowJo software.

**Cytogenetic analyses.** MEFs of each genotypes were treated with specific drugs for 12 h and released in regular media for 12 h. After 24 h, cells were arrested in metaphase stage of mitosis by using colcemid (10 μg/ml, 15210-016, KaryoMAX) for 12 h. The cells were then harvested and fixed in methanol: acetic acid (3:1). Metaphase spreads were stained with Giemsa and visualized under the microscope. Chromosomal aberrations were quantified blindly.

**Immunofluorescence.** MEFs and AFs were seeded on poly-D-Lysine coated coverslips (neuvitro GG-12) (50,000 cells) and irradiated (10 Gy followed by 3 h of recovery) or treated with camptothecin (30 nM for 24 h) or olaparib (10 nM for 24 h). Cells were treated with hypotonic solution (85.5 mM NaCl, 5 mM MgCl₂, pH 7) for 10 min followed by fixing solution (4% Paraformaldehyde, 10% SDS in PBS) for 10 min. Cells were incubated overnight at 4 °C with primary antibodies: γH2AX (1:500, Millipore JBW301), RAD51 (1:250, Millipore PC130), 53BP1 (1:1000, NovusBio NB100-304), and RPA32 (1:500, Cell Signaling Technology 2208) diluted in antibody dilution buffer (1% BSA, 0.3% TritonX100, 5% goat serum in PBS). Next morning, these were washed three times with PBS containing 0.2% TritonX 100 (PBST) and incubated with secondary antibodies (Alexa-fluor anti-mouse 594 [Invitrogen A11005], and anti-rabbit 488 [Invitrogen A11034]; 1:500) diluted in PBS at 37 °C for 1 h. The cells were washed three times with PBST and stained for DAPI (1:50,000, Sigma 11190301) for 1 min. The coverslips were mounted on clean labeled

slides with anti-fade mount (Invitrogen P36930). The slides are imaged in confocal microscope (Zeiss 710 63X) and analyzed utilizing Carl Zeiss zenblue2.6 software.

**DNA fiber assay**. In all, $5 \times 10^5$ cells were plated in 6 well plate. Pulses of 30 min each of 8 µg/ml CldU and 90 µg/ml IdU followed by 3 h 4 mM hydroxyurea (HU) treatment was performed on the cells. Fiber spreads and immunofluorescence was performed as described perviously[72]. Briefly, cells were trypsinized post treatments and resuspended in PBS. On slide cell lysis was performed by adding cell suspension and lysis buffer. After incubation for about 10 min the slides are tilted so as to fibers to spread and air dry. Methanol:acetic acid (3:1) mixture is used for fixing the fibers overnight followed by rehydration by PBS and denaturation in 2.5 M HCl for 1 h. The slides are rinsed with PBS and then blocked with 5% BSA for 40 min. After blocking the fibers are incubated with primary mouse anti-BrdU antibody (Becton Dickinson 347580, 1:500) and rat anti-BrdU antibody (Abcam ab6326, 1:500) for 2 h. Slides were rinsed with PBST and incubated with secondary anti-mouse AlexaFluor488 (1:500, Invitrogen A21202) and anti-rat AlexaFluor594 (1:500, Invitrogen A11007) for 1 h at room temperature. The slides were washed three times with PBST and were mounted (P36930, Invitrogen). The slides are imaged in Zeiss Axiocam and analyzed by Carl Zeiss zenblue2.6.

**EdU and γ-H2AX labeling in MEFs**. 50,000 WT MEF were counted and seeded on poly-D-Lysine coated coverslips (Neuvitro GG-12). The cells were irradiated (10 Gy) followed by 1 h culture in 1 mM EdU containing media. Similarly, other sets of cells were either given no treatment (control) or 30 nM/1 µM camptothecin for 1 hr along with 1 mM EdU in culture media. After 1 h the cells were permeabilized and fixed as explained above. All four sets of cells are subjected to click reaction in-situ. Briefly, the cells were given 2 PBS washes followed by 2 h incubation in click reaction solution (10 µM biotin azide, 10 mM sodium ascorbate, 2 mM CuSO₄ dissolved in PBS) at room temperature. After click reaction, the cells were washed twice with PBST and incubated overnight at 4 °C with anti-biotin antibody (Bethyl-A150-109A, 1:500) and γH2AX (1:500, Millipore JBW301) diluted in antibody dilution buffer. Rest of the steps were similar to regular immunofluorescence as described above.

**Histology**. Tissues were fixed overnight in 10% formalin, paraffin-embedded, sectioned (5 µm) and stained for H&E (hematoxylin and eosin). TUNEL staining on paraffinized sections were performed using the In Situ Cell Death Detection Kit (Roche 11684817910) following the manufacturer's directions.

**Meiotic chromosome analysis**. Meiotic spreads were prepared from the testes of 4–6-week-old male mice. The testes were briefly rinsed in PBS and then transferred to hypo-extraction buffer (30 mM Tris pH 8.2, 50 mM sucrose, 17 mM citric acid, 5 mM EDTA, 0.5 mM DTT, 0.1 mM PMSF). Gently the tunica was peeled off the testes and the tubules were teased out using fine forceps and incubated in this solution for 30 min at room temperature. On a clean labeled (prerinsed with PFA, pH 9.2 adjusted with 50 mM boric acid) slide, 25 µl of 0.1 M sucrose solution was placed and a small portion of digested tissue section from the previous step was placed. Using a fine needle this tissue was shredded and spread on the slide. These slides were kept overnight in a humid chamber to dry slowly. After the slides were ready the immunofluorescence staining was performed as above. Primary antibodies used in the study: rabbit anti-SCP3 (1:500, Abcam ab15093); mouse anti-γH2AX (1:500, Millipore JBW301); mouse anti-SCP3 (1:500 Santa Cruz sc-74568); rabbit anti-RAD51 (1:250, Millipore PC130). Secondary antibody staining was performed as described above in Immunofluorescence section.

**Streptavidin-tagged Cas9-based HR assay**. In total, $7 \times 10^5$ actively dividing PL2F7 mESCs[73] harboring 29 bp deleted blasticidin resistance gene are plated. After 24 h cells were transfected using FUGENE (Promega E2311) with 5 µg of CMV-Cas9-MSA (Addgene plasmid # 103882)[52], 5 µg U6-sgRNA/EF1a-mCherry vector cloned with blasticidin specific guideRNA (Addgene plasmid#114199)[74] and 1 µg of donor blasticidin sequence cloned in a Topo-TA vector (Invitrogen 450641) as per manufacturer's guidelines. Biotinylation of the donor plasmid was carried out using Label-IT biotin Kit (Mirus Bio 3425). After 48 h of transfection, cells were trypsinized in 5 ml Knockout DMEM + 15% FBS and plated on 100 mm culture dish with blasticidin-resistant feeders followed by 15 µg/ml blasticidin treatment for five days. After the treatment, cells were washed with PBS and fresh normal media was added until the colonies started to appear. These colonies were picked for genotyping (by Southern) and sequencing using Snapgene viewer software. Plates were stained with methylene blue (0.05% in 70% ethanol) for colony counting and assessing HR.

**Statistical analysis**. Statistical analyses were performed using Microsoft Excel and GraphPad Prism version 6.0. Exact statistical tests and $p$-values are reported for all experiments in the figure legends along with the error bars.

**Reporting summary**. Further information on research design is available in the Nature Research Reporting Summary linked to this article.

## Data availability
The data supporting the findings of this study are available from the corresponding author upon reasonable request.

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

## Acknowledgements

We thank members of our laboratory for helpful discussions and suggestions. We thank Dr. Satheesh Sengodan for his comments and critical review of the manuscript. We acknowledge the contribution of RIKEN BioResource Research Center (Ibaraki, Japan) by providing us the cryopreserved sperm of Dss1 (Sem1) knockout mice. Dr. Andre Nussenzweig for *Brca1 del 11* mutant MEFs. We thank Jennifer Harrison and Roackie Awasthi of the Cryopreservation and Assisted Reproduction Laboratory (FNLCR) for reconstituting the Dss1 knock-out mouse line. We also thank Allen Kane and Samuel Lopez (Leidos Biomedical Research, Inc., Scientific Publications, Graphics & Media Department) for help with the illustrations and text editing, respectively. This research was sponsored by the Intramural Research Program, Center for Cancer Research, National Cancer Institute, US National Institutes of Health.

## Author contributions

A.P.M. and S.A.H. performed most of the experiments and analyzed the results. R.K.C. generated the targeting construct. S.R., E.S., and L.T. performed gene targeting in ES cells and generated the knock-in mice, B.K.M. and M.A. helped with animal studies, K.K. and J.R.K. performed bone marrow transplantation studies and helped with fetal liver cell analysis. A.B.J. and B.K. performed histopathological analysis of tumors. S.S. developed

the HR assay. K.B. generated ES cells for HR assay. S.B. performed cytogenetic analysis. S.K.S. conceived the study and supervised the study. A.P.M. and S.K.S. wrote the manuscript and all authors reviewed and commented on the manuscript.

## Funding

## Competing interests
The authors declare no competing interests.
