## [Peer Review File · Nature Communications]

REVIEWER COMMENTS

Reviewer #1 (Remarks to the Author):

Review of Mishra et al

The authors present a very interesting paper on the effects of a BRCA2 missense allele that impacts homologous recombination repair (HRR) in certain situations, by virtue of disrupting interaction with an enigmatic proteosomal complex subunit, DSS1, which appears to have several functions including aiding BRCA2 to load RAD51 onto RPA-coated ssDNA at site of DSBs. The bottom line is that DSBs that are induced or arise spontaneously are not effectively repaired by HRR in BRCA2^{LP} mutants in mitotically-proliferating cells. However, the authors present evidence that these defects do not arise (or not as much) if the DSBs arise during DNA replication, and not at all during repair of meiotic DSBs. They conclude that these "exceptions" are a consequence of proximity of repair templates (homologous chromosome in the case of meiosis; sister chromatid at replication forks), which somehow overcomes the need for DSS1 to load RAD51.

Overall, the paper should be of substantial interest to the DNA repair field. There are a couple of things I'll mention that should be considered in a revision of the paper.

- 1) The writing is a bit awkward in places. Please conduct a careful round of editing.
- 2) The LP/LP genotype is semilethal. Strangely, they were present at statistically normal frequencies (just barely) at E16.5, which is just 3-4 days before birth. Were there any developmental abnormalities noted? Were the litters actually observed on the day of birth but then homozygotes died shortly after? It seems like a glaring omission not to gain some idea of the in vivo defects.
- 3) On lines 231-236, the authors talk about genomic instability in mutant cells treated with a high level of MMC. They should briefly elaborate on the nature of the chromosome aberrations, for example, do they resemble Fanconi Anemia type aberrations? Also, what are the N values listed in the legend? The number of nuclei scored or the number of different MEF lines? Hopefully the latter, in which case the N of spreads counted should be provided.
- 4) I have a couple of issues with lines 300-313. This is the section dealing with the idea that RAD51 recruitment to meiotic DSBs is normal because a repair template is close by. The premise is based on the statement "During meiotic DSB repair by HR the pairing of homologous chromosomes occurs first (during early Prophase I, leptotene stage) followed by DNA breaks generation by SPO1 (38). The cited reference is a review that doesn't even address this question. The authors need to reference primary literature to support this claim, which is the premise of their "proximity" hypothesis. I'm not sure that anyone knows, in mice, that full pairing of homologous chromosomes precedes DSB formation. Also, the authors should distinguish between synapsis and pairing, which may not be obvious to non-meiosis people. With this in mind, the statement on line 312- "This scenario perfectly mimics the meiotic DSB repair by HR (where the homologous chromosomes are already paired)" is at least an exaggeration, and

at worst totally inaccurate.

5) The low dose camptothecin trick is very clever, but it isn't entirely clear that what is being witnessed are ssBs being converted into DSBs. Are there any other possible orthogonal experiments, like genetically-induced replication errors or in vitro biochemical assays that mimic DSBs at replication forks?

{L}
{SEP:
{L}
{SEP:
{L}
{SEP:

Reviewer #2 (Remarks to the Author):

The authors describe the phenotype of mice and cells with a mutation in BRCA2, L2431P, that disables its association with DSS1. The human equivalent, L2510P, likely caused cancer in two children from one family: one with Wilms tumor and the other with T-cell acute lymphoblastic leukemia. Both LP/LP and LP/KO mice were produced and there was a reduction in their viability. LP/KO mice were small when born alive but were fertile and able to nurse in spite of reduced TEBs. They were not more susceptible to cancer, although they were hypersensitive to IR and they failed to efficiently reconstitute blood cells in lethally irradiated mice. There was also reduced ESC viability, IR-induced RAD51 foci with persistent RPAp32 foci and HR and increased GI. They crossed with the DSS-mutant mice (*dss*^{-/-} is lethal) and were unable to generate LP/KO *dss*^{-/-} mice supporting the notion that the BRCA2 mutation disrupts the DSS interaction.

The study is focus on DSB repair but BRCA2 is also important for RF maintenance. It would be interesting to see if the L2510P mutation affects RF stalls and nascent strand degradation using fiber analysis. If there is a proliferation defect, this could be the reason for the LP/LP cells failure to rescue irradiated mice with the secondary transplantation experiment.

Reviewer #3 (Remarks to the Author):

In the manuscript by Mishra et al, the investigators created a mouse model for BRCA2-L2510P, which is a patient-derived BRCA2 variant that disrupts its DSS1 interaction. The patient-derived mutation coupled with a null BRCA2 allele was analyzed and the mice were found to have decreased lifespan even though the mice were not more cancer-prone. Consistent with a FA function for the BRCA2-DSS1 interaction, these mice were deficient in hematopoietic stem and progenitor cells and have a defect in their proliferative ability. Furthermore, the BRCA2-L2510P allele is a separation-of-function allele as these mice did not exhibit fertility defects despite a defect in mitotic RAD51 focus formation upon DSB induction. The other fascinating finding is that the BRCA2-DSS1 interaction is important for DSB repair that is not replication-associated when the sister chromatid is in close proximity. Overall, this study is beautifully written and executed with proper controls throughout.

Major Comments:

1. Does blocking resection (perhaps with Mirin) rescue the defects observed with the BRCA2-L2510P mutation such as growth defects or viability?
2. To rule out if the defect is due to other replication functions, has replication fork restart or stability been analyzed in the BRCA2-L2510P mutant cells?

Minor Comments:

3. In Figure 3C, it would be helpful to have arrows pointing to where the radials are and also to label the figure as untreated and/or MMC treated. The LP/LP MMC treated radials do not show the average of 8 per nuclei, a more representative picture would be helpful.
4. In Figure 3D, it would be nice to see the distribution of the number of aberrations per nuclei rather than the average.
5. There are many typos in the discussion.

Examples:

Line 377, survive is spelled survive.

Line 393, "induces (a- needs to be added) severe HR defect"

Line 398, "Spollgenerated" should be Spo11 generated

Line 400, "We have previously reported that defect in DNA repair machinery", defect should be defects

Line 403, "as both the processes converge" should read as "as both (remove -the) processes converge on (a- needs to be added) common mechanism..."

RESPONSE TO THE REVIEWERS' COMMENTS

We are pleased to see that reviewers found our work to be of interest to the DNA repair field. We would like to thank all the reviewers for their positive and constructive comments as well as valuable suggestions. We have performed new experiments suggested by the reviewed and have made several changes in the manuscript to address their concerns. These changes are described below in our point-by-point response to their comments.

Reviewer #1:

The authors present a very interesting paper on the effects of a BRCA2 missense allele that impacts homologous recombination repair (HRR) in certain situations, by virtue of disrupting interaction with an enigmatic proteosomal complex subunit, DSS1, which appears to have several functions including aiding BRCA2 to load RAD51 onto RPA-coated ssDNA at site of DSBs. The bottom line is that DSBs that are induced or arise spontaneously are not effectively repaired by HRR in BRCA2<LP> mutants in mitotically-proliferating cells. However, the authors present evidence that these defects do not arise (or not as much) if the DSBs arise during DNA replication, and not at all during repair of meiotic DSBs. They conclude that these "exceptions" are a consequence of proximity of repair templates (homologous chromosome in the case of meiosis; sister chromatid at replication forks), which somehow overcomes the need for DSS1 to load RAD51.

Overall, the paper should be of substantial interest to the DNA repair field. There are a couple of things I'll mention that should be considered in a revision of the paper.

1) The writing is a bit awkward in places. Please conduct a careful round of editing.

We regret that the quality of our text was not satisfactory and there were several typos. We have made several changes in the text to address these concerns. In addition, the manuscript has now been edited by a professional editor to improve the writing style.

2) The LP/LP genotype is semilethal. Strangely, they were present at statistically normal frequencies (just barely) at E16.5, which is just 3-4 days before birth. Were there any developmental abnormalities noted? Were the litters actually observed on the day of birth but then homozygotes died shortly after? It seems like a glaring omission not to gain some idea of the in vivo defects.

Similar to the *LP/KO* hemizygous mice, *LP/LP* homozygous genotype is also semilethal. However, the reduction in numbers is not as severe as observed for *LP/KO* mice. The phenotypes observed in *LP/LP* mice, embryos, hematopoietic stem and progenitor cells, MEFs as well as adult fibroblasts are all milder than in *LP/KO*.

As suggested by the reviewer, we have now carefully characterized the *LP/LP* embryos to determine the time of their lethality and also examined the embryos for any developmental

defects. At 16.5dpc, their numbers are reduced but the reduction is marginal. Other than a reduction in the size of some *LP/LP* embryos, we did not notice any apparent developmental defects in them at 16.5dpc. We have also performed histopathological analysis but observed no defects. We have shown these results in Suppl. Fig 2a and c.

When we examined the embryos at 18.5dpc, we obtained *LP/LP* embryos at expected Mendelian ratios ($p=0.75$). Some *LP/LP* embryos were clearly smaller in size, but we observed no other abnormalities (see Supp. Figure 2b). Interestingly, when we genotyped the newborn pups, we found the number of *LP/LP* pups to be significantly reduced (obtained 11 pups instead of the expect 23.75, $p=0.0002$, please see revised Table 2). It is possible that some of the *LP/LP* embryos were still born and/or were eaten immediately by their mothers. We predict that the smaller embryos (at 16.5dpc and 18.5dpc) were unable to survive as we did not notice any difference in the size of the *LP/LP* pups at birth or weaning. Their growth (Figure 1c, d) was indistinguishable from their control littermates. Also, we did not observe loss of *LP/LP* mice after birth. We have now described these observations on *LP/LP* embryos in the manuscript (page 6, 2nd paragraph, last three sentences and Page 7, 1st paragraph) and provided the number of embryos of various genotypes at 18.5dpc and at birth in the revised Table 2. At present we do not fully understand why some *LP/LP* embryos fail to survive. Since the focus of this study is on the role of DSS1-BRCA2 interaction on RAD51 recruitment, we did not investigate the cause of the lethality of *LP/LP* embryos in detail.

3) On lines 231-236, the authors talk about genomic instability in mutant cells treated with a high level of MMC. They should briefly elaborate on the nature of the chromosome aberrations, for example, do they resemble Fanconi Anemia type aberrations? Also, what are the N values listed in the legend? The number of nuclei scored or the number of different MEF lines? Hopefully the latter, in which case the N of spreads counted should be provided.

We have now described the kind of chromosomal aberrations we observed in MEFs treated with MMC. We have also added a new Suppl. Figure 3a, which describes the individual aberrations we observed. We have used three independent MEFs of each genotype. We have now clarified this and also indicated the number of spreads examined per MEF in the figure legends.

4) I have a couple of issues with lines 300-313. This is the section dealing with the idea that RAD51 recruitment to meiotic DSBs is normal because a repair template is close by. The premise is based on the statement "During meiotic DSB repair by HR the pairing of homologous chromosomes occurs first (during early Prophase I, leptotene stage) followed by DNA breaks generation by SPO1 (38). The cited reference is a review that doesn't even address this question. The authors need to reference primary literature to support this claim, which is the premise of their "proximity" hypothesis. I'm not sure that anyone knows, in mice, that full pairing of homologous chromosomes precedes DSB formation. Also, the authors should distinguish

between synapsis and pairing, which may not be obvious to non-meiosis people. With this in mind, the statement on line 312- "This scenario perfectly mimics the meiotic DSB repair by HR (where the homologous chromosomes are already paired)" is at least an exaggeration, and at worst totally inaccurate.

We sincerely regret the errors in our manuscript as described by the reviewer. We unintentionally made a mistake in describing some of the key steps of meiotic recombination and did not explain the order of events correctly. We agree that formation of double strand breaks is essential for synapsis of homologous chromosome and also full pairing of homologous chromosomes does not occur prior to DSB formation. As SPO11 generated DSBs are repaired by HR, recombination between homologous chromosomes promotes their synapsis and is marked by the formation of synaptonemal complex.

In addition to the synapsis of homologous chromosomes that is DSB-dependent, spatial alignment of homologous chromosomes has been reported during the pre-leptotene stage of Prophase I. It occurs between the telomere of homologous chromosomes and is referred to as the bouquet stage. At this stage, the homologous chromosomes are not fully paired but they are present in close proximity. This partial pairing has been shown to be independent of Spo11 and is observed in many organisms including mice. We believe that the presence of homologous chromosomes in close proximity may facilitate displacement of RPA and loading of RAD51 in the absence of DSS1 bound to BRCA2 in *LP/LP* and *LP/KO* spermatocytes. We have now described these events and explained our hypothesis correctly (page 15, 2nd paragraph).

5) The low dose camptothecin trick is very clever, but it isn't entirely clear that what is being witnessed are ssBs being converted into DSBs. Are there any other possible orthogonal experiments, like genetically-induced replication errors or in vitro biochemical assays that mimic DSBs at replication forks?

PARP proteins are known to be required for repair of DNA single strand breaks and PARP inhibitors inhibit repair of these breaks that get converted into DSBs. Therefore, to provide an additional evidence to support our hypothesis, we examined whether PARPi treatment can restore RAD51 recruitment in *LP/LP* and *LP/KO* MEFs. Similar to camptothecin treatment, 10nM Olaparib treatment was able to generate RAD51 foci in mutant cells. While 10nM Olaparib was sufficient to increase chromosomal aberrations in MEFs deficient in BRCA1 (*Brca1 del ex11*), we did not observe an increase in chromosomal aberrations in *LP/LP* and *LP/KO* MEFs suggesting that the RAD51 foci formation leads to repair of the DSBs. We have described these findings in the manuscript on Page 16 , 2nd and 3rd paragraphs) and included these results in Figure 5d, e and f and Suppl. Figure 7d.

Reviewer #2:

*The authors describe the phenotype of mice and cells with a mutation in BRCA2, L2431P, that disables its association with DSS1. The human equivalent, L2510P, likely caused cancer in two children from one family: one with Wilms tumor and the other with T-cell acute lymphoblastic leukemia. Both LP/LP and LP/KO mice were produced and there was a reduction in their viability. LP/KO mice were small when born alive but were fertile and able to nurse in spite of reduced TEBs. They were not more susceptible to cancer, although they were hypersensitive to IR and they failed to efficiently reconstitute blood cells in lethally irradiated mice. There was also reduced ESC viability, IR-induced RAD51 foci with persistent RPAp32 foci and HR and increased GI. They crossed with the DSS-mutant mice (*dss*^{-/-} is lethal) and were unable to generate LP/KO *dss*^{-/-} mice supporting the notion that the BRCA2 mutation disrupts the DSS interaction.*

The study is focus on DSB repair but BRCA2 is also important for RF maintenance. It would be interesting to see if the L2510P mutation affects RF stalls and nascent strand degradation using fiber analysis. If there is a proliferation defect, this could be the reason for the LP/LP cells failure to rescue irradiated mice with the secondary transplantation experiment.

As suggested by the reviewer, we have now examined the impact of L2510P/L2431P variant on protection of stalled replication forks. We performed DNA fiber assay using *WT*, *KO*+, *LP*+, *LP/LP* and *LP/KO* MEFs and used *Brcal* mutant MEFs (*Brcal* ^{$\Delta 11/\Delta 11$}) as a control for unprotected replication forks. As expected, we observed defect in fork protection in *Brcal* mutant MEFs, none of the other MEFs displayed any defect in fork protection suggesting that L2510P/L2431P variant has no effect on the integrity of stalled forks. We have described these results on Page 13, 1st paragraph, and showed the DNA fiber assay data in Suppl. Fig 3b.

Reviewer #3:

In the manuscript by Mishra et al, the investigators created a mouse model for BRCA2-L2510P, which is a patient-derived BRCA2 variant that disrupts its DSS1 interaction. The patient-derived mutation coupled with a null BRCA2 allele was analyzed and the mice were found to have decreased lifespan even though the mice were not more cancer-prone. Consistent with a FA function for the BRCA2-DSS1 interaction, these mice were deficient in hematopoietic stem and progenitor cells and have a defect in their proliferative ability. Furthermore, the BRCA2-L2510P allele is a separation-of-function allele as these mice did not exhibit fertility defects despite a defect in mitotic RAD51 focus formation upon DSB induction. The other fascinating finding is that the BRCA2-DSS1 interaction is important for DSB repair that is not replication-associated when the sister chromatid is in close proximity. Overall, this study is beautifully written and executed with proper controls throughout.

Major Comments:

1. Does blocking resection (perhaps with Mirin) rescue the defects observed with the BRCA2-L2510P mutation such as growth defects or viability?

As suggested by the reviewer, we examined the cause of growth defect observed in *LP/LP* and *LP/KO* MEFs by treating the cells with mirin. Considering the role of MRE11 in resection of DSBs to initiate repair of DSBs by HR, it is possible that blocking resection may overcome the cell cycle stalling if it is caused by a defect in HR-mediated DNA repair. To test this, we treated *WT*, *LP/LP* and *LP/KO* MEFs with 10 μ M and 25 μ M mirin and monitored their proliferation for 7 days and counted the cell numbers.

Figure 1: Impact of MRE11 inhibition on proliferation of MEFs when treated (t) with 10 μ M (left) and 25 μ M mirin (right) relative to untreated (ut) MEFs.

As shown above in Figure 1 (left), there was no difference in the proliferation of untreated MEFs and those treated with 10 μ M mirin. When we increased the concentration of mirin to 25 μ M (Figure 1, right), although *LP/KO* MEFs had no effect, *WT* and *LP/LP* MEFs exhibited a decline in proliferation, likely due to a defect in MRE11-mediated DNA repair, which is not apparent in the *LP/KO* MEFs. Because the results are inconclusive, we have not included it in the manuscript.

2. To rule out if the defect is due to other replication functions, has replication fork restart or stability been analyzed in the BRCA2-L2510P mutant cells?

We have now examined the impact of L2510P/L2431P variant on protection of stalled replication forks. We performed DNA fiber assay using *WT*, *KO/+*, *LP/+*, *LP/LP* and *LP/KO* MEFs and used *Brca1* mutant MEFs (*Brca1^{Δ11/Δ11}*) as a control for unprotected replication forks. As expected, we observed defect in fork protection in *Brca1* mutant MEFs, none of the other MEFs displayed any defect in fork protection suggesting that L2510P/L2431P variant has no effect on the integrity of stalled forks. We have described these results on Page 13, 1st paragraph, and showed the DNA fiber assay data in Suppl. Fig 3b.

Minor Comments:

3. In Figure 3C, it would be helpful to have arrows pointing to where the radials are and also to label the figure as untreated and/or MMC treated. The LP/LP MMC treated radials do not show the average of 8 per nuclei, a more representative picture would be helpful. need to fix images.

As suggested by the reviewer, we have marked all the chromosomal aberrations with an arrow and circled the radials in Figure 3c. We have labelled the panels as “treated” and “untreated”. We have now included images that better represent the average number of aberrations observed in the MEFs of each genotype.

4. In Figure 3D, it would be nice to see the distribution of the number of aberrations per nuclei rather than the average.

We have revised the figure (new Figure 3d) to show the distribution of aberrations per nuclei. Also, we have now included the types of chromosomal aberrations observed in the treated and untreated MEFs in Suppl. Figure 3a.

5. There are many typos in the discussion.

We sincerely regret these errors. We have carefully edited the manuscript and corrected all the typos including those listed below.

Examples:

Line 377, survive is spelled survive.

Line 393, “induces (a- needs to be added) severe HR defect”

Line 398, “Spollgenerated” should be Spo11 generated

Line 400, “We have previously reported that defect in DNA repair machinery”, defect should be defects

Line 403, “as both the processes converge” should read as “as both (remove -the) processes converge on (a- needs to be added) common mechanism.

REVIEWERS' COMMENTS

Reviewer #1 (Remarks to the Author):

The authors have done an outstanding job in revising the manuscript. They actually performed several new experiments that enhanced the paper and reinforced conclusions. Well done, nice findings.

John Schimenti

Reviewer #3 (Remarks to the Author):

The authors have now addressed my concerns.